# Using Both Demonstrations and Language Instructions to Efficiently Learn Robotic Tasks

**Albert Yu**
UT Austin
`albertyu@utexas.edu`

**Raymond J. Mooney**
UT Austin
`mooney@utexas.edu`

## Abstract

Demonstrations and natural language instructions are two common ways to specify and teach robots novel tasks. However, for many complex tasks, a demonstration or language instruction alone contains ambiguities, preventing tasks from being specified clearly. In such cases, a combination of *both* a demonstration *and* an instruction more concisely and effectively conveys the task to the robot than either modality alone. To instantiate this problem setting, we train a single multi-task policy on a few hundred challenging robotic pick-and-place tasks and propose DeL-TaCo (Joint Demo-Language Task Conditioning), a method for conditioning a robotic policy on task embeddings comprised of two components: a visual demonstration and a language instruction. By allowing these two modalities to mutually disambiguate and clarify each other during novel task specification, DeL-TaCo (1) substantially decreases the teacher effort needed to specify a new task and (2) achieves better generalization performance on novel objects and instructions over previous task-conditioning methods. To our knowledge, this is the first work to show that simultaneously conditioning a multi-task robotic manipulation policy on *both* demonstration and language embeddings improves sample efficiency and generalization over conditioning on either modality alone. See additional materials at `https://deltaco-robot.github.io`.

## 1 Introduction

A significant barrier to deploying household robots is the inability of novice users to teach new tasks with minimal time and effort. Recent work in multi-task learning suggests that training on a wide range of tasks, instead of the single target task, helps the robot learn shared perceptual representations across the different tasks, improving generalization (Kalashnikov et al., 2021; Yu et al., 2019; Jang et al., 2021; Shridhar et al., 2021). We study the problem of how to more efficiently specify new tasks for multi-task robotic policies while also improving performance.

Humans often learn complex tasks through multiple concurrent modalities, such as simultaneous visual and linguistic (speech/captioning) streams of a video tutorial. One might reasonably expect robotic policies to also benefit from multi-modal task specification. However, previous work in multitask policies condition only on a single modality during evaluation: one-hot embeddings, language embeddings, or demonstration/goal-image embeddings. Each has limitations.

One-hot encodings for each task (Kalashnikov et al., 2021; Ebert et al., 2021) suffice for learning a repertoire of training tasks but perform very poorly on novel tasks where the one-hot embedding is out of the training distribution, since one-hot embedding spaces do not leverage semantic similarity between tasks to more rapidly learn additional tasks. Conditioning policies on goal-images (Nair et al., 2017; 2018; Nasiriany et al., 2019) or training on video demonstrations (Smith et al., 2020; Young et al., 2020) often suffer from ambiguity, especially when there are large differences between the environment of the demonstration and the environment the robot is in, hindering the understanding of a demonstration's true intention. In language-conditioned policies (Blukis et al., 2018; 2019; Mees et al., 2021; 2022), issues of ambiguity are often even more pronounced, since humans specify similar tasks in very linguistically dissimilar ways and often speak at different levels of granularity, skipping over common-sense steps and details while bringing up other impertinent information. Grounding novel nouns and verbs not seen during training compounds these challenges.

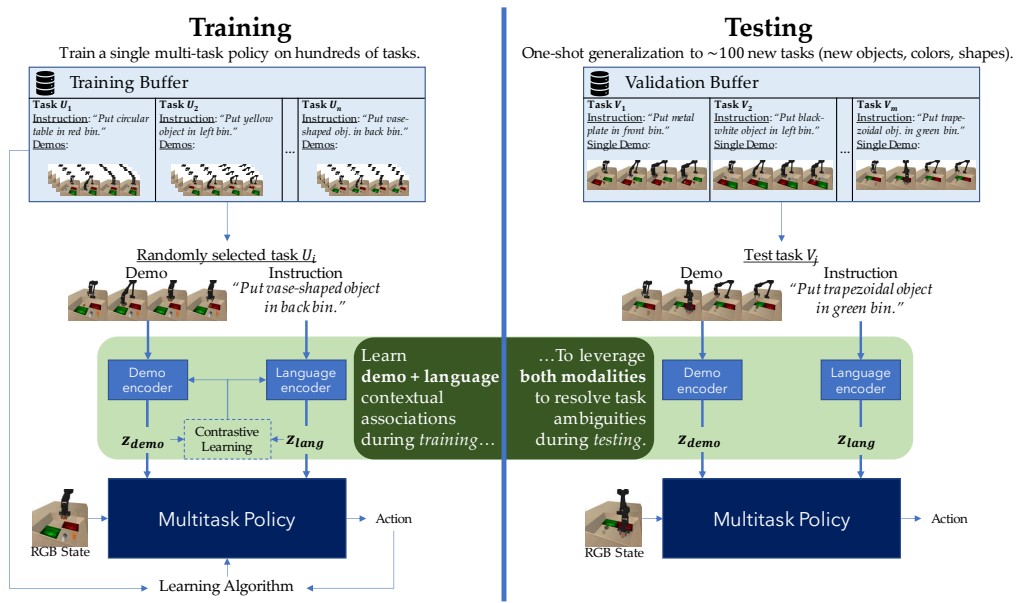

Figure 1: **DeL-TaCo Overview.** Unlike current multitask methods that condition on a single task specification modality, DeL-TaCo simultaneously conditions on both language and demonstrations during training and testing to resolve any ambiguities in either task specification modality, enabling better generalization to novel tasks and significantly reducing teacher effort for specifying new tasks.

We posit that in a broad category of tasks, current unimodal task representations are often too inefficient and ambiguous for novel task specification. In these tasks, current task-conditioning methods would need either a large number of diverse demonstrations to disambiguate the intended task, or a long, very detailed, fine-grained language instruction. Both are difficult for novice users to provide. We argue that conditioning the policy on *both* a demonstration *and* language not only ameliorates the ambiguity issues with language-only and demonstration-only specifications, but is *much easier and more cost-effective for the end-user to provide*.

We propose DeL-TaCo (Figure 1), a new task embedding scheme comprised of two component modalities that contextually complement each other: demonstrations of the target task and corresponding language descriptions. To our knowledge, this is the first work to demonstrate that specifying new tasks to robotic multi-task policies simultaneously with both demonstrations and language reduces teacher effort in task specification and improves generalization performance, two important characteristics of deployable household robots. With bimodal task embeddings, ambiguity is bidirectionally resolved: instructions disambiguate intent in demonstrations, and demonstrations help ground novel noun and verb tokens by conveying what to act on, and how. To learn several hundred tasks, we train a single imitation learning (IL) policy, conditioned on joint demonstration-language embeddings, to predict low-level continuous-space actions for a robot given image observations. Task encoders are trained jointly with the policy, making our model fully differentiable end-to-end.

To summarize, our main contributions are as follows: (1) We present a broad distribution of highly-randomized simulated robotic pick-and-place tasks where instructions or demonstrations alone are too ambiguous and inefficient at specifying novel tasks. (2) We propose a simple architecture, DeL-TaCo, for training and integrating demonstrations and language into joint task embeddings for few-shot novel task specification. This framework is flexible and learning algorithm-agnostic. (3) We show that DeL-TaCo significantly lowers teacher effort in novel task-specification and improves generalization performance over previous unimodal task-conditioning methods.

## 2 RELATED WORK

### 2.1 MULTI-TASK LEARNING

The most straightforward way to condition multi-task policies is through one-hot vectors (Ebert et al., 2021; Kalashnikov et al., 2021; Walke et al., 2022; Yu et al., 2021). We instead use embed-

ding spaces that are shaped with pretrained language models so that semantically similar tasks are encoded in similar regions of the embedding space, which helps improve generalization. Multi-task robotic policies have also been studied in other settings and contexts that do not fall under the class of approaches we take in this paper, such as hierarchical goal-conditioned policies (Gupta et al., 2022), probabilistic modeling techniques (Wilson et al., 2007), distillation and transfer learning (Parisotto et al., 2015; Teh et al., 2017; Xu et al., 2020; Rusu et al., 2015), data sharing (Espeholt et al., 2018; Hessel et al., 2019), gradient-based techniques (Yu et al., 2020), policy modularization (Andreas et al., 2017; Devin et al., 2017) and task modularization (Yang et al., 2020).

## 2.2 LEARNING WITH LANGUAGE AND DEMONSTRATIONS

**Conditioning Multitask Policies on Language or Demonstrations.** Our work largely tackles the same problem as BC-Z (Jang et al., 2021) of generalizing to novel tasks with multi-task learning. BC-Z trains a video demonstration encoder to predict the pretrained embeddings of corresponding language instructions, while jointly training a multi-task imitation learning policy conditioned on *either* the instruction *or* demonstration embeddings. Lynch & Sermanet (2021); Mees et al. (2021) learn a similar policy conditioned on either language or goal images. All of these approaches learn to map a demonstration or goal image to a similar embedding space as its corresponding language instruction. During training, Mees et al. (2022) use both demonstrations and language to learn associations between demonstration embeddings and language-conditioned latent plans, but during evaluation, only use the language embedding to produce a latent plan. With a slightly different architecture, Shao et al. (2020) learn a policy that maps natural language verbs and initial observations to full trajectories by training a video classifier on a large dataset of annotated human videos.

While all of these prior approaches use both demonstrations and language during training, their policies are conditioned on *either* a language instruction *or* visual image/demonstration embedding during testing. By contrast, ours is conditioned on *both* demonstration *and* language embeddings during training and testing, which we show improves generalization performance and reduces human teacher effort on a broad category of tasks.

**Pretrained Multi-modal Models for Multitask Policies.** Another recent line of work leverages pretrained vision-language models to learn richer vision features for downstream policies. CLIPort (Shridhar et al., 2021) uses pre-trained CLIP (Radford et al., 2021) to learn robust Transporter-based (Zeng et al., 2020) robot policies. Our method resembles CLIPort, its 3-dimensional successor PerAct (Shridhar et al., 2022), and the previously mentioned multi-task policy methods in that we train on expert trajectories associated with language task descriptions, but in CLIPort and PerAct, the policy is *only conditioned on language* during training and testing; demonstrations are used only as buffer data for imitation learning. Our method, however, learns tasks during training or testing by using *both language and a demonstration* to condition the policy.

ZeST (Cui et al., 2022) and Socratic Models (Zeng et al., 2022) demonstrate that pretrained vision-language models encode valuable information for robotic goal selection and task specification. R3M (Nair et al., 2022) pretrains a ResNet (He et al., 2015) policy backbone on language-annotated videos from Ego4D (Grauman et al., 2021) to boost downstream task performance. While our motivation is similar to ZeST in using a pretrained language model to leverage the structure of the pretrained embedding space, we assume access to both language and demonstrations for learning novel tasks and condition on task embeddings from both, which is unlike the ZeST and R3M problem settings where the policies are not directly task-conditioned.

## 2.3 OTHER APPLICATIONS OF LANGUAGE FOR ROBOTICS

**Language-shaped state representations.** On the MetaWorld multitask benchmark (Yu et al., 2019), a number of prior works have investigated using language instructions to learn better state representations. Sodhani et al. (2021) use language instruction embeddings to compute an attention-weighted context representation over a mixture of state encoders. Silva et al. (2021) learn a goal encoder that transforms language instruction embeddings to shape the state encoder representations.

**Hierarchical Learning with Language.** Our approach can be loosely framed as hierarchical learning, where we have two high-level task encoders that output language and demonstration embeddings, both of which the low-level policy is conditioned on to output actions. Prior work has used

language instructions in hierarchical learning for shaping high-level plan vectors (Mees et al., 2022) or skill representations (Garg et al., 2022), which are then fed to a low-level policy to output the action. Karamcheti et al. (2021) use an autoencoder-based architecture to predict higher-dimensional robot actions from lower-dimensional controller actions and language instructions, where the language is fed into both the encoder and decoder. All of these prior approaches condition on a single high-level policy, whereas ours incorporates guidance from two high-level encoders for *both* demonstrations and language to learn novel tasks, giving the low-level policy access to certain information expressible only through their combination.

**Language for Rewards and Planning.** Language has also been used for reward shaping in RL (Nair et al., 2021; Goyal et al., 2019; 2020). Pretrained language models have also been leveraged for their ability to propose plans in long-horizon tasks (Huang et al., 2022; Ahn et al., 2022; Chen et al., 2022). While we work with IL instead of RL and mainly deal with highly variable pick-and-place tasks, we do not use language for training reward functions or for planning, though our multi-modal task specification framework is compatible with these additional uses of language.

## 3 PROBLEM SETTING

### 3.1 MULTI-TASK IMITATION LEARNING

We define a set of $n$ tasks $\{T_i\}_{i=1}^{n}$ and split them into training tasks $U$ and test tasks $V$, where $(U, V)$ is a bipartition of $\{T_i\}_{i=1}^{n}$. For each task $T_i$, we assume access to a set of $m$ expert trajectories $\{\tau_{ij}\}_{j=1}^{m}$ and a single language description $l_i$. Given continuous state space $\mathcal{S}$, continuous action space $\mathcal{A}$, and task embedding space $\mathcal{Z}$, the goal is to train a Markovian policy $\pi : \mathcal{S} \times \mathcal{Z} \to \Pi(\mathcal{A})$ that maps the current state and task embeddings to a probability distribution over the continuous action space.

During training, we assume access to a buffer $\mathcal{D}_{\text{train}}$ of trajectories for only the tasks in $U$ and their associated natural language descriptions. We define each trajectory as a fixed-length sequence of state-action pairs $\tau_{ij} = \left[ \left( s_{0,j}^{(i)}, a_{0,j}^{(i)} \right), \left( s_{1,j}^{(i)}, a_{1,j}^{(i)} \right), ... \right]$, where $j$ is the trajectory index for task $T_i \in U$ with task embedding $z_i$. We use behavioral cloning (BC) (Hussein et al., 2017; Pomerleau, 1988) to update the parameters of $\pi$ to maximize the log probability of $\pi \left( a_{t,j}^{(i)} \big| s_{t,j}^{(i)}, z_i \right)$, though our framework is agnostic to the learning algorithm and would work for RL approaches as well.

During evaluation, we assume access to a buffer $\mathcal{D}_{\text{val}}$ of trajectories for only the tasks in $V$ and their associated natural language descriptions. Unlike $\mathcal{D}_{\text{train}}$ where we have $m$ demonstrations for each task, in $\mathcal{D}_{\text{val}}$ we have just a single demonstration for each task. For all test tasks $T_i \in V$, we rollout the policy for a fixed number of timesteps by taking action $a_t \sim \pi(a|s_t, z_i)$. The $z_i$ for all test tasks is computed beforehand and held constant throughout each test trajectory.

### 3.2 TASK ENCODER NETWORKS

To obtain the task embedding $z_i$, we have two encoders (which are either trained jointly with policy $\pi$ or frozen from a pretrained model): a *demonstration encoder*, $f_{demo} : \tau_{ij} \mapsto z_{demo,i}$ mapping trajectories of task $T_i$ to demonstration embeddings, and a *language encoder*, $f_{lang} : l_i \mapsto z_{lang,i}$ mapping task instruction strings $l_i$ to language embeddings. Previous work has explored using $z_i$ as a one-hot task vector, language embedding $z_{lang,i}$, *or* goal image/demonstration embedding $z_{demo,i}$, but our approach DeL-TaCo uses task embedding $z_i = [z_{demo,i}, z_{lang,i}]$ based on *both* the instruction *and* demonstration embedding during training and testing to learn novel tasks.

## 4 METHOD

### 4.1 ARCHITECTURE

**Demonstration and Language Encoders.** The encoder $f_{demo}$ is a CNN network trained from scratch. Following Jang et al. (2021), we input the demonstration as an array of $m \times n$ frames (in raster-scan order) from the trajectory for faster processing. (We use $(m, n) = (1, 2)$ or $(2, 2)$ in our

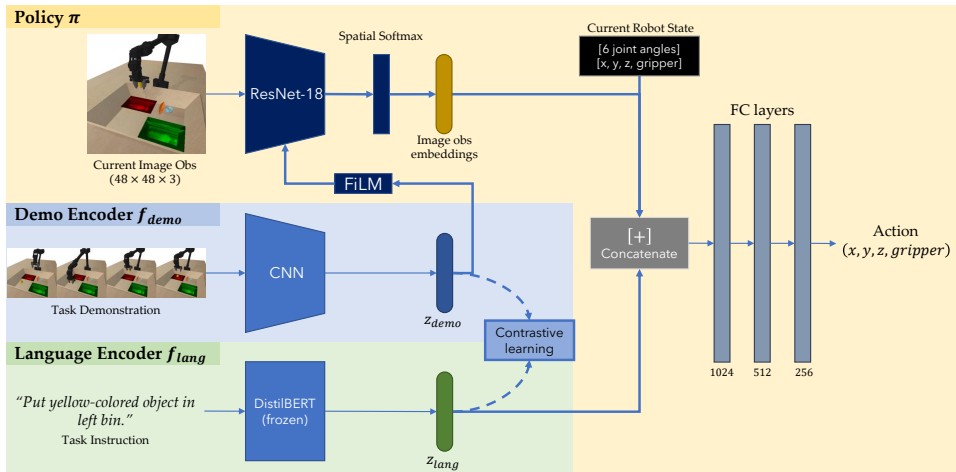

Figure 2: **Method Architecture.** DeL-TaCo uses three main networks: the policy $\pi$, a demonstration encoder $f_{demo}$, and a language encoder $f_{lang}$. During both training and testing, the policy is conditioned on the demonstration and language embeddings for the task.

experiments.) We freeze a pretrained DistilBERT (Sanh et al., 2019) as the encoder $f_{lang}$, where $z_{lang,i}$ is simply the average of all DistilBERT-embedded tokens in $l_i$ (we found this works better than taking the [CLS] token embedding).

**Policy Network.** We use a ResNet-18 (He et al., 2015) as the visual backbone for the policy $\pi$, followed by a spatial softmax layer (Finn et al., 2016) and fully connected layers.

**Task Conditioning Architecture.** BC-Z (Jang et al., 2021) inputs the task embedding into the ResNet backbone via FiLM (Perez et al., 2018) layers, which apply a learned affine transformation to the intermediate image representations after each residual block. BC-Z's task embeddings are either from demonstrations *or* language. Since our policy conditions on both, the main architectural decision was finding the best way to feed task embeddings from multiple modalities into the policy.

Empirically, a simple approach performed best. The demonstration embeddings $z_{demo}$ are fed into the policy's ResNet backbone via FiLM, while the language task embeddings $z_{lang}$ and robot proprioceptive state (6 joint angles, end-effector xyz coordinates, and gripper open/close state) are concatenated to the output of the spatial softmax layer. Our full network architecture is shown in Figure 2, hyperparameters are in Appendix B, and architectural ablations are in Table 8.

## 4.2 TRAINING AND LOSSES

The training procedure for DeL-TaCo is summarized in Algorithm 1. During each training iteration, we sample a size $k$ subset of training tasks $M = \{T_{m_1}, ..., T_{m_k}\} \subset U$. Given a trajectory $\tau_{ij}$ for task $T_{m_i}$ and corresponding natural language instruction $l_i$, we compute the demonstration embeddings $z_{emb,m_i} = f_{demo}(\tau_{ij})$ and language embeddings $z_{lang,m_i} = f_{lang}(l_i)$. We collect the embeddings of tasks in $M$ in matrices $Z_{demo} = [z_{demo,m_1}, ..., z_{demo,m_k}]$ and $Z_{lang} = [z_{lang,m_1}, ..., z_{lang,m_k}]$.

To train the demonstration encoder, Jang et al. (2021) use a cosine distance loss to directly regress demonstration embeddings to their associated language embeddings. However, this causes demonstration embeddings to be essentially equivalent to the associated language embeddings for each task, undercutting the value of passing both to our policy. To preserve information unique to each modality while enabling the language and demonstration embedding spaces to shape each other, we train with a CLIP-style (Radford et al., 2021) contrastive loss for our demonstration encoder:

$$\mathcal{L}_{demo}(Z_{demo}, Z_{lang}) = CrossEntropy\left(\frac{1}{\beta}Z_{demo}^{\top}Z_{lang}, I\right) \tag{1}$$

where $I$ is the identity matrix and $\beta$ is a tuned temperature scalar. We use the standard BC log-likelihood loss as the policy loss term for some trajectory composed of state-action pairs $x_{t,i,j} =$

$\left(s_{t,j}^{(i)}, a_{t,j}^{(i)}\right)$ extracted from an expert demonstration $\tau_{ij}$ for task $T_{m_i}$:

$$\mathcal{L}_{policy}(\tau_{ij}) = -\sum_{x_{t,i,j} \in \tau_{ij}} \log \pi\left(a_{t,j}^{(i)} \big| s_{t,j}^{(i)}, z_{demo,m_i}, z_{lang,m_i}\right) \tag{2}$$

Both $f_{demo}$ and $\pi$ networks are trained jointly with the following loss, for a tuned $\alpha > 0$:

$$\mathcal{L}(\pi, f_{demo}, f_{lang}) = \mathcal{L}_{policy}(\tau_{ij}) + \alpha\mathcal{L}_{demo}(Z_{demo}, Z_{lang}) \tag{3}$$

Where $\mathcal{L}_{policy}(\tau_{ij})$ is summed over all trajectories in the batch of training tasks $M$ (we omit this double summation in Equation 2 for brevity). Note that the language encoder does not have a loss term because we use a frozen, pretrained language model and rely on the pretrained embedding space to shape the demonstration encoding space.

### 4.3 EVALUATION

During evaluation, we want the robot to perform some novel task $T_v \in V$. Recall that $T_v \notin U$, our set of training tasks. From our problem setup description in Section 3.1, we have access to a validation task buffer $\mathcal{D}_{\text{val}}$ with a single demonstration $\tau_v$ and a natural language instruction $l_v$ of task $T_v$. We encode the demonstration with $f_{demo}$ and the language with $f_{lang}$ and pass both task embeddings to the policy. Details are summarized in Algorithm 2.

| **Algorithm 1** DeL-TaCo: Training | **Algorithm 2** DeL-TaCo: Evaluation |
|---|---|
| **Input:** $\mathcal{D}_{\text{train}}$ | **Input:** $\mathcal{D}_{\text{val}}$ |
| 1: **while not** done **do** | 1: **for** validation task $T_v$ in $V$ **do** |
| 2: $\quad M \leftarrow k$ random train tasks from $U$ | 2: $\quad$ Get 1 demo $\tau_v$ and language $l_v$ from $\mathcal{D}_{\text{val}}$ |
| 3: $\quad$ Sample $X_i = \{\tau_{ij}\}_{j=0}^{b-1} \sim \mathcal{D}_{\text{train}}$ | 3: $\quad z_{demo} \leftarrow f_{demo}(\tau_v)$ // Encode demo |
| 4: $\quad X \leftarrow \{X_i | T_i \in M\}$ | 4: $\quad z_{lang} \leftarrow f_{lang}(l_v)$ // Encode language |
| 5: $\quad L \leftarrow \{l_i | T_i \in M\}$ // Lang. instructions | 5: $\quad$ **for** time $t = 0, ..., H-1$ **do** |
| 6: $\quad Z_{demo} \leftarrow f_{demo}(X)$ // Demo encoder | 6: $\quad\quad$ Take action $a_t \sim \pi(a|s_t, z_{demo}, z_{lang})$ |
| 7: $\quad Z_{lang} \leftarrow f_{lang}(L)$ // Language encoder | 7: $\quad$ **end for** |
| 8: $\quad$ Update $\pi, f_{demo}$ on $\mathcal{L}(\pi, f_{demo}, f_{lang})$ // per Eqn. 3 | 8: **end for** |
| 9: **end while** | |

## 5 EXPERIMENTS

We empirically investigate the following questions: (1) Does there exist a distribution of tasks that are more clearly specified with both language and demonstrations rather than either alone? (2) Does conditioning on both language instructions and video demonstrations with DeL-TaCo improve generalization performance on novel tasks? (3) If so, how much teacher effort is reduced by specifying a new task with both language and demonstrations than with either modality alone?

### 5.1 SETUP

**Environment.** We develop a Pybullet (Coumans & Bai, 2007-2022) simulation environment with a WidowX 250 robot arm, 32 possible objects of diverse colors and shapes for manipulation, and 2 different containers. The action space is continuous, representing an $(x, y, z)$ change in the robot's end effector position, plus the binary gripper state (closed/opened). We subdivide the workspace into four quadrants. Two quadrants are randomly chosen to contain the two different containers, and three of the 32 possible objects are dropped at random locations in the remaining two quadrants. RGB image observations are size $48 \times 48 \times 3$ and fed into the policy. The input format of each demo for $f_{demo}$ is an $m \times n$ array of images extracted from the trajectory. Details are in Appendix H.

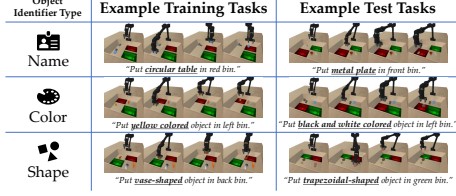

Figure 3: **Sample train and test tasks**, grouped by the object identifier types (underlined in each language instruction). All 6 container identifiers are seen in both training and testing.

**Task Objective.** To explore the first question, we design the following set of pick-and-place tasks where the objective is to grasp the target object and place it in the target container. Both the target object and container can be inferred from the demonstration and language instruction. In every task, the scene contains three visually distinct objects (of which exactly one of them is the target object) and two visually distinct containers (of which exactly one of them is the target container). To make the task more challenging, the distractor objects on the scene are chosen adversarially, whenever possible, to match either the color or shape of the target object. Thus, a robotic policy that disregards both the task demonstration and instruction and picks any random object and places it into any random container would solve the task with 1-in-6 odds.

**Language Instructions for Each Task.** Figure 3 shows a selection of our training and testing tasks. Each task is specified through language with a single template-based instruction of the format "put [target object identifier] in [target container identifier]."

We make this environment more challenging by having task instructions refer to containers by either their color or quadrant position and objects by either their name, color, or shape. We use six different container identifiers in the task instructions to convey which container to drop the object in: red, green, front, back, left, and right. Thus, if the robot is provided a demonstration of grasping a cup and placing it in the red container in the front left quadrant, and it encounters an initialization with the containers in different locations, it does not know whether to place the cup in the red, front, or left container. This ambiguity can only be resolved with the language instruction. Conversely, aspects of the task, such as which object to grasp, are most clearly expressed through the demonstration rather than the instruction, since for novel tasks, the language instruction contains object identifiers unfamiliar to the policy. The task instructions also refer to the objects through different types of identifiers: their unique names (32 strings such as "fountain vase"), color (8 strings such as "black-and-white colored object"), or shape (10 strings such as "trapezoidal prism shaped object").

The multiple identifiers help simulate ambiguity that arises from informal human instructions, where different humans may refer to the same object or container through different attributes, enabling demonstrations and instructions to complement each other when the robot learns a new task. In total, there are 50 target object identifiers ($32 + 8 + 10$) and 6 target container identifiers, giving us 300 pick-and-place tasks. We train and evaluate on a bipartition of these 300 tasks. See Appendix A for a list of all our train and test tasks.

**Success Metric.** In calculating the success rate, a successful trajectory is defined as one that (1) picks up the correct object *and* (2) places it in the correct container. Appendix D details the number of seeds and trials used to calculate success rates and standard deviations.

**Data.** Using a scripted policy (details in Appendix C), we collect roughly 130 successful demonstrations for each training task, and a single successful demonstration for each test task. All demonstrations are 30 timesteps long. Depending on our experimental scenario (see Section 5.2), we train on $65\%$ to $80\%$ of the 300 tasks, so our training buffer contains roughly 26,000-31,000 trajectories.

## 5.2 GENERALIZATION PERFORMANCE ON NOVEL TASKS

To test generalization, we run experiments under two scenarios: (A) generalization to novel objects, colors and shapes, and (B) generalization to only novel colors and shapes.

### 5.2.1 SCENARIO A: NOVEL OBJECTS, COLORS, AND SHAPES

Table 1 (plots in Figure 8) shows our results in experimental scenario A, where we train on 24/32 objects, 4/8 colors, and 5/10 shapes (a total of 198 training tasks) and evaluate on the remaining 102 tasks. The overall success rates show much room for improvement on this challenging benchmark for a number of reasons. The robot must not only know how to pick-and-place the 8/32 objects it has never seen during training, but must also understand novel instructions that refer to these objects by either their name, color, or shape. Additionally, training a multitask policy to perform well on hundreds of tasks remains an open question with current multitask robot learning algorithms—a problem compounded in difficulty by the adversarial selection of distractor objects in our environment as mentioned earlier in Section 5.1.

We lower-bound the performance of our task conditioning methods by first running a one-hot conditioned policy, with the expectation that it performs worse than conditioning on language and/or demonstrations for the reasons mentioned in Section 1. As an upper-bound, we directly train a

Table 1: Evaluation on Novel Objects, Colors, and Shapes. (p) = pretrained.

| Demo Encoder | Language Encoder | Task Conditioning | Success Rate ± SD (%) |
|---|---|---|---|
| – | – | One-hot (lower bound) | 6.6 ± 1.3 |
| – | – | One-hot Oracle (upper bound) | 47.9 ± 4.8 |
| | CLIP (p) | Language-only | 13.7 ± 1.9 |
| | | Demo-only | 8.0 ± 1.9 |
| | | **DeL-TaCo (ours)** | **15.3 ± 1.8** |
| – | DistilBERT (p) | Language-only | 10.4 ± 1.6 |
| CNN | – | Demo-only | 14.6 ± 2.2 |
| CNN | DistilBERT (p) | **DeL-TaCo (ours)** | **19.9 ± 1.8** |
| CNN | – | BC-Z (Jang et al., 2021); Demo-only | 8.8 ± 2.0 |
| CNN | DistilBERT (p) | MCIL (Lynch & Sermanet, 2021); Demo-only + Language-only | 9.4 ± 1.8 |

Table 2: Evaluation on Novel Colors and Shapes. (p) = pretrained.

| Demo Encoder | Language Encoder | Task Conditioning | Success Rate ± SD (%) |
|---|---|---|---|
| – | – | One-hot (lower bound) | 10.3 ± 1.8 |
| – | – | One-hot Oracle (upper bound) | 50.9 ± 4.9 |
| – | DistilBERT (p) | Language-only | 15.8 ± 2.8 |
| CNN | – | Demo-only | 17.0 ± 2.7 |
| CNN | DistilBERT (p) | **DeL-TaCo (ours)** | **26.3 ± 4.1** |

one-hot oracle on only the 102 evaluation tasks and evaluate on those same tasks. No other method in the table is trained on any evaluation tasks. (For consistency, the one-hot oracle is trained on the same total number of trajectories as the other methods.)

Next, we examine the performance of policies conditioned with only language, with only one demonstration, and with both (DeL-TaCo). The language-only policies do not involve training $f_{demo}$, and only the language instruction embeddings are fed into the policy via FiLM during training and testing. The demo-only policies train $f_{demo}$ as shown in Algorithm 1, but during training and testing, only the demonstration embedding $z_{demo}$ is passed into the policy via FiLM. DeL-TaCo (ours) conditions on *both* demonstration *and* language during training and testing as shown in Algorithms 1 and 2.

When using pretrained DistilBERT as $f_{lang}$ and a lightweight CNN for $f_{demo}$, DeL-TaCo achieves the highest performance, increasing the success rate of the second-best conditioning method, demo-only, from $14.6\%$ to $19.9\%$. Both methods using demonstration embeddings outperform the language-conditioned policy perhaps because a visual demonstration is important in conveying the nature of the chosen object and how the robot should manipulate it. Note that both the demo-only and DeL-TaCo policies train the $f_{demo}$ CNN from scratch without any pretraining, so they must learn to ground object and container identifiers from training demonstrations alone.

We also compare to prior methods. BC-Z (Jang et al., 2021) performs worse than our approaches because its demo encoder is trained to directly regress $z_{demo}$ to $z_{lang}$, hindering it from performing better than solely using $z_{lang}$ during testing. MCIL (Lynch & Sermanet, 2021), which trains separate encoders for each task embedding modality and averages the imitation learning losses over the different encoders, also performs worse than DeL-TaCo because without any task encoder loss term, it is harder to learn a well-shaped task embedding space, hurting generalization performance.

Finally, to evaluate the effect of pretraining, we use pretrained CLIP (Radford et al., 2021) as the task encoder (with its language transformer as $f_{lang}$ and vision transformer as $f_{demo}$) and freeze it during training. The language-only policy performs significantly better than the video-only policy most likely because CLIP's visual transformer was trained mostly on real-world images and without further finetuning, does not know how to sufficiently differentiate between the simulation demonstrations of different tasks in our problem setting. Despite this, DeL-TaCo modestly outperforms conditioning on language-only or demo-only, demonstrating the value of our method even with frozen pretrained models.

### 5.2.2 SCENARIO B: NOVEL COLORS AND SHAPES

In Table 2 (plots in Figure 9), we train on 32/32 objects, 4/8 colors, and 5/10 shapes, and evaluate on the rest—an easier setting as all objects were seen during training. Since evaluation tasks in this scenario only refer to objects by their color or shape, we up-sample the color and shape training tasks to be 50% of each training batch (such up-sampling was not done in scenario A).

We take the highest-performing $f_{demo}$ and $f_{lang}$ models in Table 1 and again compare conditioning on language, demonstrations, and both. All methods perform better in scenario B than A. The novel color and shape task demonstrations contain more ambiguity than the novel object demonstrations because the task with language instruction "put the blue object in the left bin" might have a demon-

Table 3: Value of Language. Evaluation on Novel Objects, Colors, and Shapes.

| Task Conditioning | Demo-only | | | | | DeL-TaCo (ours) |
|---|---|---|---|---|---|---|
| # demos per test-task finetuned on | 0 | 10 | 25 | 50 | 100 | 0 |
| Success Rate (%) | 14.6 | 14.9 | 17.4 | 20.0 | 24.2 | 19.9 |
| ± SD (%) | ±2.2 | ±1.6 | ±2.7 | ±2.4 | ±2.5 | ±1.8 |

stration where the robot manipulates the blue cup, but the test-time environment might contain a blue table instead. This added ambiguity likely explains the increased importance of language and the wider $9.7\%$ performance gap between DeL-TaCo and demo-only task conditioning.

**Analysis.** Overall, we see that on this wide range of tasks, language and demonstrations together do help disambiguate each other during task specification—answering our first question; this leads to better generalization performance on novel tasks—answering our second question.

### 5.3 How many demonstrations is language worth?

To answer our third question, we re-examine experimental scenario A (testing on novel objects, colors, and shapes) but now further finetune the demo-only policy on a variable number of test-task expert demonstrations. Results are shown in Table 3 (plots in Figure 9). The demo-only policy only starts to match and surpass DeL-TaCo (underlined) when it is finetuned on 50 demonstrations (underlined) *per evaluation task* (a total of around 5,000 demonstrations for all test tasks combined). This suggests that surprisingly, specifying a new task to DeL-TaCo with a single demonstration and language instruction performs as well as specifying a new task to a demo-only policy with a single demonstration *and finetuning it on 50 additional demonstrations of that task*. This showcases the immense value of language in supplementing demonstrations for novel task specification, significantly reducing the effort involved in teaching robots novel tasks over demonstration-only methods.

### 5.4 Ablations and Multimodality for Ambiguity Resolution

We provide extensive ablations of our model and algorithm in Appendix F and further explore the utility of using multi-modal task specification to resolve ambiguity in Appendix G.

## 6 Conclusion

When specifying tasks through language or demonstrations, ambiguities can arise that hinder robot learning, especially when the demonstrations or instructions were provided in an environment that does not perfectly align with the environment the robot is in. In this paper, we showed a problem setting of learning 300 highly diverse pick and place tasks and propose a simple framework, DeL-TaCo, to resolve ambiguity during task specification by using both language and demonstrations during both training and testing. Two main obstacles to deploying household robotic systems are the inability to generalize to new environments and tasks, and prohibitively high end-user effort needed to teach robots these new tasks. Our results show progress on both fronts: over previous task-conditioning methods, DeL-TaCo improves generalization performance to new tasks by $5-9\%$ and reduces human effort on our set of tasks by roughly 50 expert demonstrations per task.

**Limitations and Future Work.** Our work leaves a number of areas for improvement. First, we experiment only with pick-and-place tasks. Future work may need more interpretable modular encoders to handle a wider diversity of manipulation skills and temporally-extended tasks. Second, we used a rigid set of template-based language instructions for each task, but our framework would likely benefit from a more diverse instruction set of human paraphrases for each task. Third, we did not find pretrained vision-language models, such as CLIP, to increase performance in our simulation-based environment, most likely because of the domain mismatch between our simulation objects and the more real-world-centric images CLIP was trained on. Investigating ways to better leverage pretrained vision-language models for multimodal task specification, in tandem with real-world robotic tasks and real-world human demonstrations, would be a promising line of future research.

## 7 REPRODUCIBILITY STATEMENT

Please see our appendix for details needed to replicate our results. In particular, Appendix A provides the full list of our 300 tasks, instructions, and objects, as well as train and test task splits. Appendix B contains architectural hyperparameters and details for network layers, initialization, and optimizer settings. The remaining appendices detail aspects of our training/evaluation processes and provide additional ablations that were not fully described in the main text of our paper.

We link to our open-sourced codebase on our project website, `https://deltaco-robot.github.io`.

## 8 ETHICS STATEMENT

This work leverages language model embeddings for task conditioning, which leaves our approach vulnerable to distributional biases of the language models. Future work should explore adding additional safeguards to DeL-TaCo to reason about the safety of language instructions before utilizing these embeddings for execution by the robot policy.

### ACKNOWLEDGMENTS

We would like to thank Prasoon Goyal and Vanya Cohen for their numerous suggestions and discussions during the course of this work. This research was supported by NSF NRI Grant IIS-1925082.

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

# Appendices

## A  ALL TASKS, INSTRUCTIONS, AND TRAIN-TEST SPLIT

### A.1  LIST OF TASKS

All 300 tasks are shown below, by object identifier (rows) and container identifier (columns). The colors denote groups of tasks which guide our train and test task splits, and the cell numbers denote the task indices.

- Scenario A (novel objects, colors, and shapes) trains on all gray tasks and tests on yellow, blue, and green tasks.

- Scenario B (novel colors and shapes) trains on all gray and yellow tasks and tests on blue and green tasks.

| Object Identifier Type | Object Identifier | Container Identifier | | | | | |
|---|---|---|---|---|---|---|---|
| | | green | red | front | back | left | right |
| name | conic cup | 0 | 50 | 100 | 150 | 200 | 250 |
| | fountain vase | 1 | 51 | 101 | 151 | 201 | 251 |
| | circular table | 2 | 52 | 102 | 152 | 202 | 252 |
| | hex deep bowl | 3 | 53 | 103 | 153 | 203 | 253 |
| | smushed dumbbell | 4 | 54 | 104 | 154 | 204 | 254 |
| | square prism bin | 5 | 55 | 105 | 155 | 205 | 255 |
| | narrow tray | 6 | 56 | 106 | 156 | 206 | 256 |
| | colunnade top | 7 | 57 | 107 | 157 | 207 | 257 |
| | stalagcite chunk | 8 | 58 | 108 | 158 | 208 | 258 |
| | bongo drum bowl | 9 | 59 | 109 | 159 | 209 | 259 |
| | pacifier vase | 10 | 60 | 110 | 160 | 210 | 260 |
| | beehive funnel | 11 | 61 | 111 | 161 | 211 | 261 |
| | crooked lid trash can | 12 | 62 | 112 | 162 | 212 | 262 |
| | toilet bowl | 13 | 63 | 113 | 163 | 213 | 263 |
| | pepsi bottle | 14 | 64 | 114 | 164 | 214 | 264 |
| | tongue chair | 15 | 65 | 115 | 165 | 215 | 265 |
| | modern canoe | 16 | 66 | 116 | 166 | 216 | 266 |
| | pear ringed vase | 17 | 67 | 117 | 167 | 217 | 267 |
| | short handle cup | 18 | 68 | 118 | 168 | 218 | 268 |
| | bullet vase | 19 | 69 | 119 | 169 | 219 | 269 |
| | glass half gallon | 20 | 70 | 120 | 170 | 220 | 270 |
| | flat bottom sack vase | 21 | 71 | 121 | 171 | 221 | 271 |
| | trapezoidal bin | 22 | 72 | 122 | 172 | 222 | 272 |
| | vintage canoe | 23 | 73 | 123 | 173 | 223 | 273 |
| | bathtub | 24 | 74 | 124 | 174 | 224 | 274 |
| | flowery half donut | 25 | 75 | 125 | 175 | 225 | 275 |
| | t cup | 26 | 76 | 126 | 176 | 226 | 276 |
| | cookie circular lidless tin | 27 | 77 | 127 | 177 | 227 | 277 |
| | box sofa | 28 | 78 | 128 | 178 | 228 | 278 |
| | two layered lampshade | 29 | 79 | 129 | 179 | 229 | 279 |
| | conic bin | 30 | 80 | 130 | 180 | 230 | 280 |
| | jar | 31 | 81 | 131 | 181 | 231 | 281 |
| color | black and white | 32 | 82 | 132 | 182 | 232 | 282 |
| | brown | 33 | 83 | 133 | 183 | 233 | 283 |
| | blue | 34 | 84 | 134 | 184 | 234 | 284 |
| | gray | 35 | 85 | 135 | 185 | 235 | 285 |
| | white | 36 | 86 | 136 | 186 | 236 | 286 |
| | red | 37 | 87 | 137 | 187 | 237 | 287 |
| | orange | 38 | 88 | 138 | 188 | 238 | 288 |
| | yellow | 39 | 89 | 139 | 189 | 239 | 289 |
| shape | vase | 40 | 90 | 140 | 190 | 240 | 290 |
| | chalice | 41 | 91 | 141 | 191 | 241 | 291 |
| | freeform | 42 | 92 | 142 | 192 | 242 | 292 |
| | bottle | 43 | 93 | 143 | 193 | 243 | 293 |
| | canoe | 44 | 94 | 144 | 194 | 244 | 294 |
| | cup | 45 | 95 | 145 | 195 | 245 | 295 |
| | bowl | 46 | 96 | 146 | 196 | 246 | 296 |
| | trapezoidal prism | 47 | 97 | 147 | 197 | 247 | 297 |
| | cylinder | 48 | 98 | 148 | 198 | 248 | 298 |
| | round hole | 49 | 99 | 149 | 199 | 249 | 299 |

### A.2  TASK INSTRUCTION FORMAT

As mentioned in Section 5.1, we use the following template as the language instruction for each task: "Put [target object identifier string] in [target container identifier string]." For each object identifier, we build a string referring to the target obj in a specific format shown in Table 4.

Table 4: **Object Identifier String Format for each Object Identifier Type.**

| Object Identifier Type | Target Object Identifier String |
|---|---|
| Name | "[object color] colored, [object shape] shaped [object name]" |
| Color | "[object color] colored object" |
| Shape | "[object shape] shaped object" |

Example task instructions (with target object identifier and target container strings underlined):

- Task 4: "Put black and white colored, chalice shaped smushed dumbbell in green bin."
- Task 292: "Put cup shaped object in right bin."

### A.3 TRAIN AND TEST SPLIT VISUALIZATIONS

We visually show our train-test splits on objects (Figure 4), colors (Figure 5), and shapes (Figure 6).

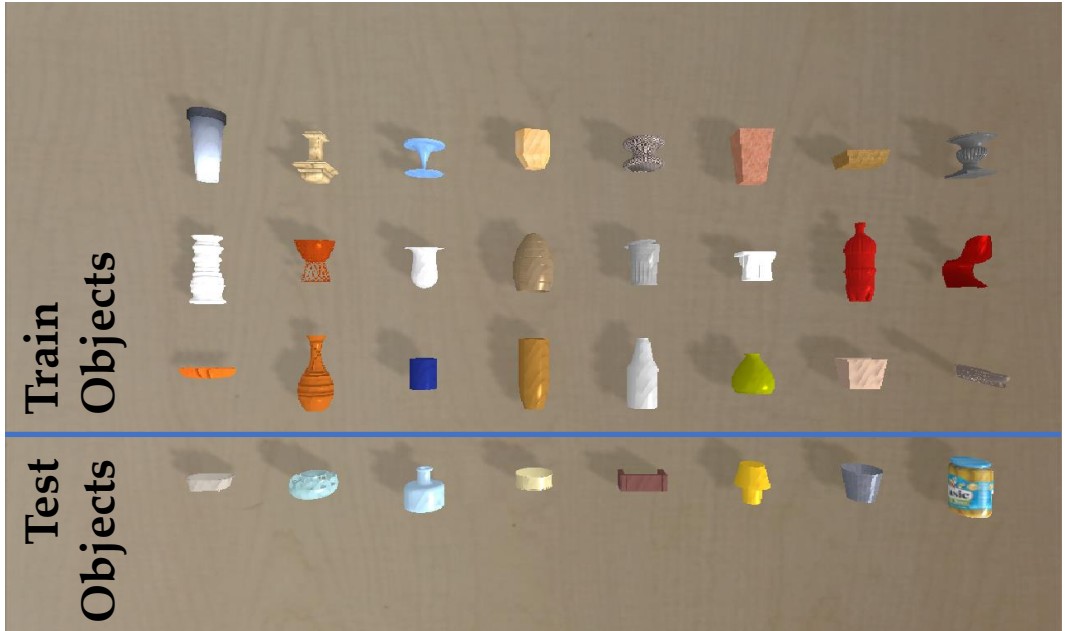

Figure 4: **Train-Test Object Split**. Objects are shown in raster-scan task-index order, so the object in the second row from top, second column from left, is the "bongo drum bowl", which is associated with task index 9.

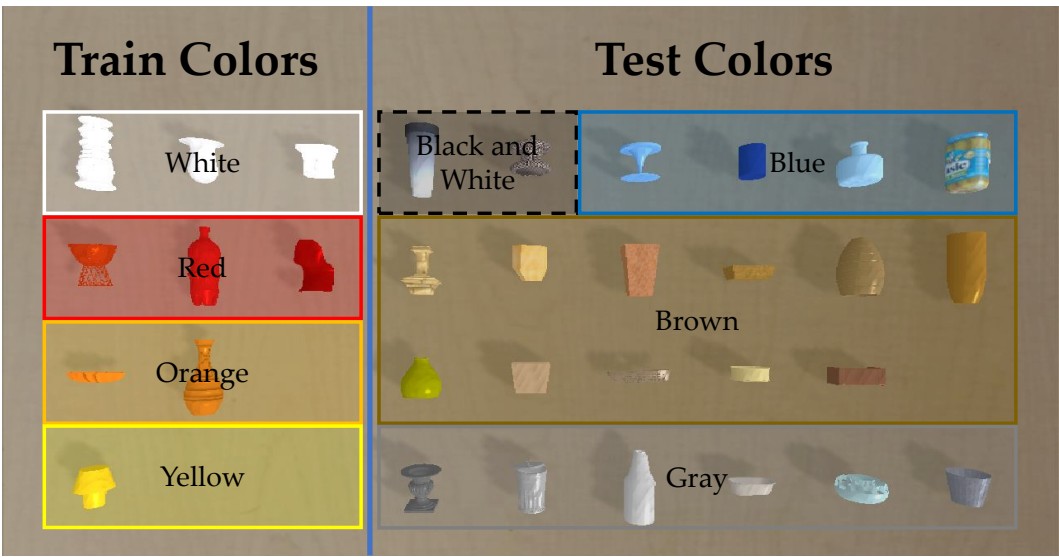

Figure 5: **Train-Test Color Split.**

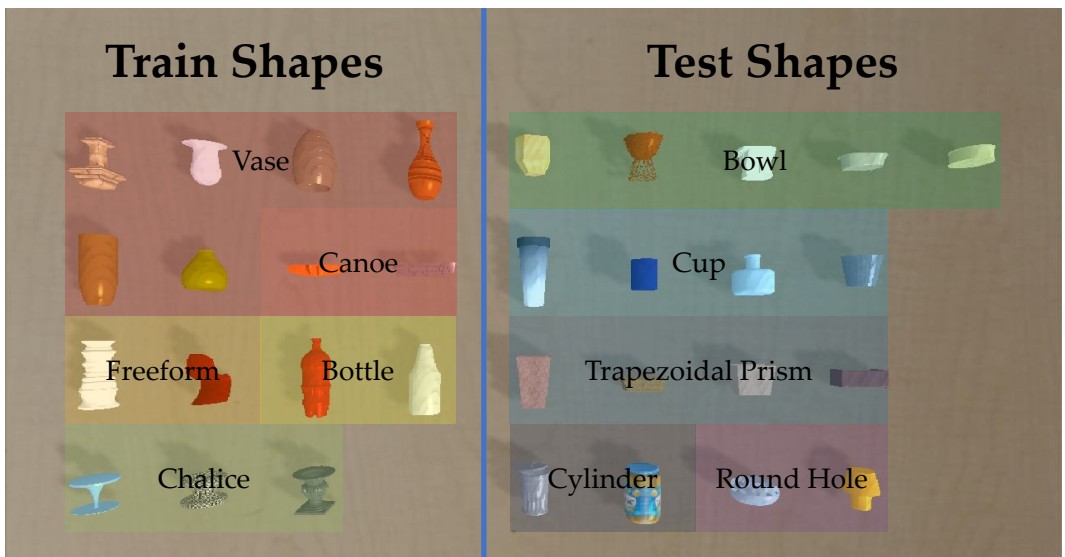

Figure 6: **Train-Test Shape Split.**

## B  DETAILED ARCHITECTURE AND HYPERPARAMETERS

### B.1  POLICY $\pi$ AND DEMONSTRATION ENCODER $f_{demo}$ ARCHITECTURE

See Figure 7 for a detailed diagram of the policy and demonstration encoder (for a higher-level overview, see Figure 2). For the policy backbone, we use a ResNet-18 architecture but made changes to the strides and number of channels to adapt the network to our small image size. Hyperparameters are shown in Tables 5 and 6.

### B.2  TRAINING HYPERPARAMETERS

Table 7 shows our IL training hyperparameters.

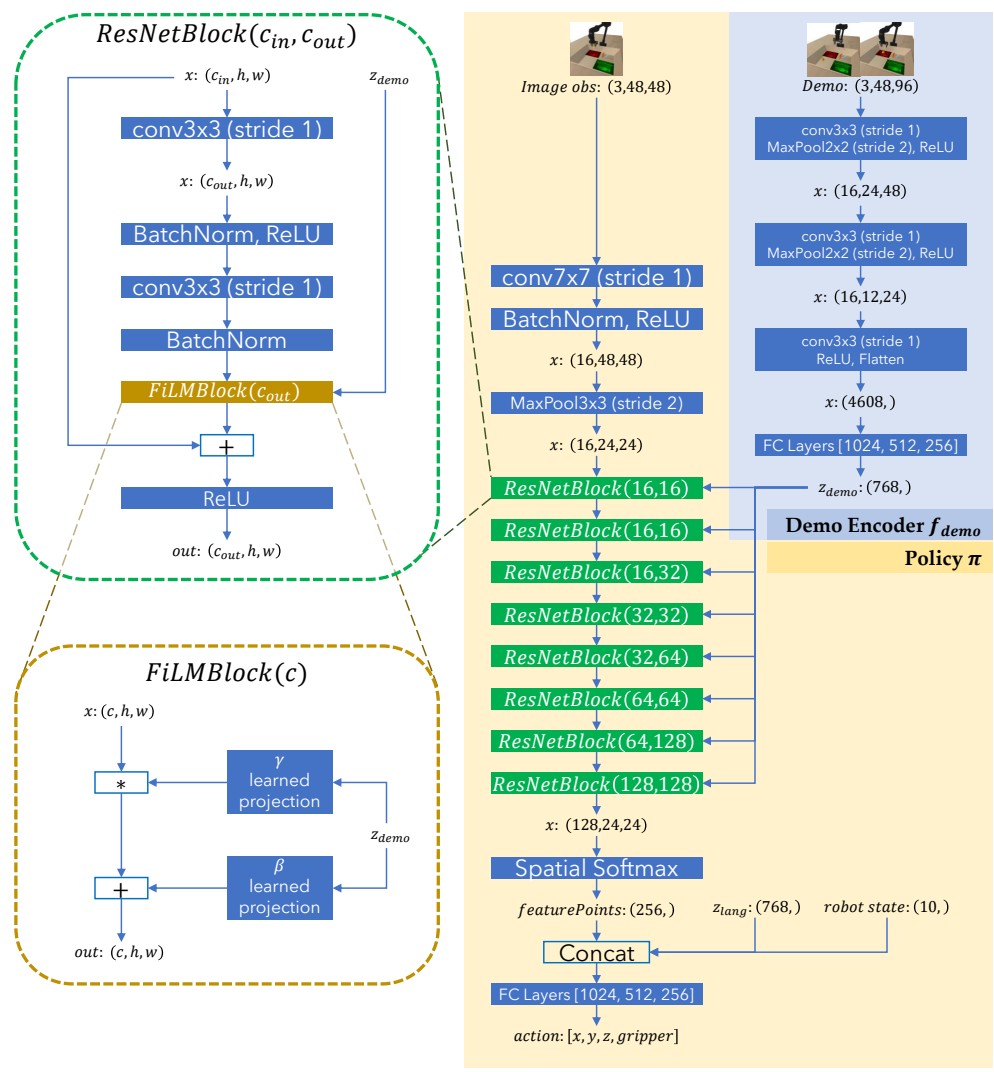

Figure 7: **Detailed Architecture of the Policy and Demonstration Encoder.**

Table 7: **Imitation learning hyperparameters.** In each training iteration, we sample 16 random tasks from our training buffer and get 64 samples for each task, for a total batch size of 1024.

| Attribute | Value |
|---|---:|
| Number of Tasks per Batch | 16 |
| Batch Size per Task | 64 |
| Learning Rate | $3 \times 10^{-4}$ |
| Task Encoder weight ($\alpha$ in $\mathcal{L}$) | 10.0 |
| Contrastive Learning Temperature ($\beta$ in $\mathcal{L}_{demo}$) | 0.1 |

Table 5: **Policy $\pi$ hyperparameters.**

| Attribute | Value |
|---|---|
| Input Height | 48 |
| Input Width | 48 |
| Input Channels | 3 |
| Number of Kernels | [16, 32, 64, 128] |
| Kernel Sizes | [7, 3, 3, 3, 3] |
| Conv Strides | [1, 1, 1, 1, 1] |
| Maxpool Stride | 2 |
| Fully Connected Layers | [1024, 512, 256] |
| Hidden Activations | ReLU |
| FiLM input size | 768 |
| FiLM hidden layers | 0 |
| Spatial Softmax Temperature | 1.0 |
| Learning Rate | $3 \times 10^{-4}$ |
| Policy Action Distribution | Multivariate Isotropic Gaussian $\mathcal{N}(\mu, \sigma)$ |
| Policy Outputs | $(\mu, \sigma)$ |
| Image Augmentation | Random Crops |
| Image Augmentation Padding | 4 |

Table 6: $f_{demo}$ **CNN hyperparameters.**

| Attribute | Value |
|---|---|
| Demonstration frames | First and last timesteps |
| Demonstration image array size $(m, n)$ | $(1, 2)$ |
| Input Height ($m \cdot$ Image height) | 48 |
| Input Width ($n \cdot$ Image width) | 96 |
| Input Channels | 3 |
| Output Size | 768 |
| Kernel Sizes | [3, 3, 3] |
| Number of Kernels | [16, 16, 16] |
| Strides | [1, 1, 1] |
| Fully Connected Layers | [1024, 512, 256] |
| Hidden Activations | ReLU |
| Paddings | [1, 1, 1] |
| Pool Type | Max 2D |
| Pool Sizes | [2, 2, 1] |
| Pool Strides | [2, 2, 1] |
| Pool Paddings | [0, 0, 0] |
| Image Augmentation | Random Crops |
| Image Augmentation Padding | 4 |

## C  SCRIPTED POLICY DETAILS

We collect $26,000 - 31,000$ training demonstrations using a scripted policy with Gaussian noise added to the action of each timestep. Details are shown in Algorithm 3. "eePos" stands for end-effector position. All variables ending in "Pos" are xyz positions. Our training buffer contains only successful demonstrations.

---

**Algorithm 3** Scripted Pick and Place

---

1: pickPos ← target object position
2: dropPos ← target container position
3: distThresh ← 0.02
4: numTimesteps ← 30
5: placeAttempted ← False
6: **for** t **in** [0, numTimesteps) **do**
7:     eePos ← end effector position
8:     dropPosDist ← $\|$eePos − dropPos$\|_2$
9:     pickPosDist ← $\|$eePos − pickPos$\|_2$
10:     **if** placeAttempted **then**
11:         action ← 0
12:     **else if** object not grasped **AND** pickPosDist > distThresh **then**
13:         // Move toward target object
14:         action ← pickPos − eePos
15:     **else if** object not grasped **then**
16:         // gripper is very close to object
17:         action ← pickPos − eePos
18:         close gripper  // Object is in gripper
19:     **else if** object not lifted **then**
20:         // Move gripper upward to avoid hitting other objects/containers
21:         action ← [0, 0, 1]
22:     **else if** dropPosDist > distThresh **then**
23:         // Move toward target container
24:         action ← dropPos − eePos
25:     **else**
26:         open gripper  // Object falls into container
27:         placeAttempted ← True
28:     **end if**
29:     noise ∼ $\mathcal{N}(0, 0.1)$
30:     action ← action + noise
31:     $s' ←$ env.step(action)
32: **end for**

---

## D    SUCCESS RATE CALCULATION DETAILS

To avoid reporting cherry-picked results, we detail our success rate calculation methodology here.

We run each setting with three random seeds for 800k-900k training steps. An evaluation set, which we define as rolling out the policy for 2 trials per task for all of the test tasks, is run every 10k training steps. Thus, there are a total of 80-90 evaluation sets that occur throughout training. Let seed $i$ attain the success rate $r(i, j)$ on evaluation set $j$. Let $J =$ top 10 evaluation set indices for the quantity $\text{mean}_i r(i, j)$. Our reported success rate and standard deviation in the tables are calculated as the following equations:

$$\text{Reported Success Rate} = \text{mean}_{j \in J}(\text{mean}_i r(i, j))$$
$$\text{Reported Standard Deviation} = \text{mean}_{j \in J}(\text{stddev}_i r(i, j))$$

Scenario A: Since there are 102 test tasks, each success rate in Tables 1 and 3 is computed from:

$$\frac{2 \text{ trials}}{\text{test task}} \times \frac{102 \text{ test tasks}}{\text{evaluation set}} \times \frac{10 \text{ evaluation sets}}{\text{seed}} \times 3 \text{ seeds} = 6120 \text{ evaluation trials}$$

Scenario B: Applying the same calculation for the 54 tasks in Scenario B, each success rate in Table 2 is computed from 3240 evaluation trials.

For Table 3, the best final checkpoint of the three demo-only policy seeds from Table 1 was taken for finetuning.

# E   LEARNING CURVES FOR EXPERIMENTS

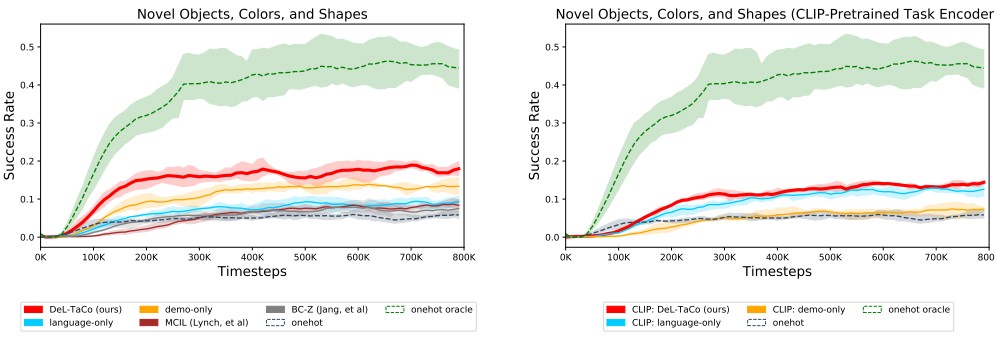

Figure 8: Table 1 learning curves, where all methods are evaluated on novel objects, colors, and shapes. *Left:* $f_{demo}$ is a trained-from-scratch CNN and $f_{lang}$ is pretrained DistilBERT. *Right:* $f_{demo}$ and $f_{lang}$ are from pretrained CLIP. The same upper and lower one-hot bounds (dotted) are shown in both the left and right plots.

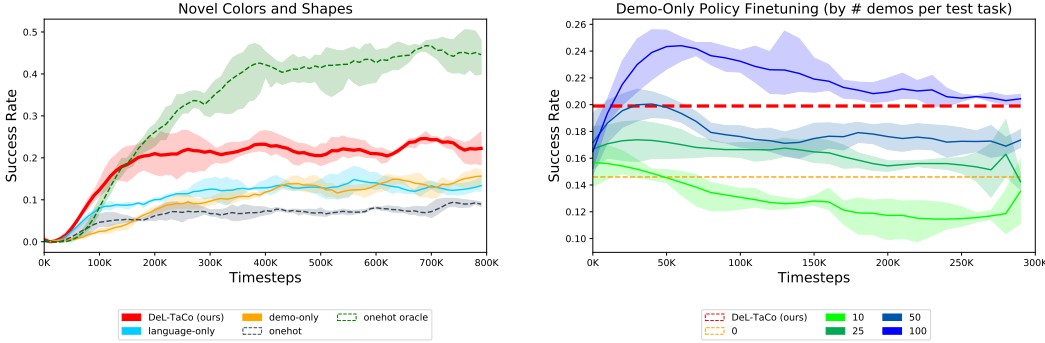

Figure 9: Table 2 and 3 learning curves. *Left:* Evaluation only on novel colors and shapes. *Right:* Evaluation on novel objects, colors, and shapes, using a trained-from-scratch CNN as $f_{demo}$ and pretrained DistilBERT as $f_{lang}$. The performance of the demo-only policy and DeL-TaCo policy from Table 1 (also depicted in the left plot of Figure 8) are shown as lower and upper dotted lines, respectively. The solid lines indicate performance during 300k finetuning steps when given $x$ demonstrations per test-task, where $x$ is indicated in the legend.

Table 8: Ablations. Evaluation on Novel Objects, Colors and Shapes. (p) = pretrained.

| Demo Encoder | Language Encoder | Task Encoder Loss | Task Conditioning Arch. | Task Conditioning | Success Rate ± SD (%) |
|---|---|---|---|---|---|
| Best non-oracle result from Table 1 | | | | | |
| CNN | DistilBERT (p) | Contrastive | FiLM (demo), Concat (lang) | DeL-TaCo (ours) | 19.9 ± 1.8 |
| Language Encoder Ablations | | | | | |
| – | DistilBERT (finetuned) | – | FiLM | Language-only | 12.6 ± 5.0 |
| CNN | DistilBERT (finetuned) | Contrastive | FiLM (demo), Concat (lang) | DeL-TaCo (ours) | 12.1 ± 5.6 |
| – | DistilRoBERTa (p) | – | FiLM | Language-only | 12.4 ± 2.1 |
| CNN | DistilRoBERTa (p) | Contrastive | FiLM (demo), Concat (lang) | DeL-TaCo (ours) | 16.6 ± 1.5 |
| – | miniLM (p) | – | FiLM | Language-only | 13.7 ± 2.3 |
| CNN | miniLM (p) | Contrastive | FiLM (demo), Concat (lang) | DeL-TaCo (ours) | **20.9 ± 2.3** |
| CLIP (p) | CLIP (p) + MLP head | Contrastive | FiLM (lang), Concat (demo) | DeL-TaCo (ours) | 15.8 ± 2.3 |
| Demo Encoder Ablations | | | | | |
| R3M (p) + MLP head | DistilBERT (p) | Contrastive | FiLM (demo), Concat (lang) | DeL-TaCo (ours) | 13.2 ± 1.8 |
| Task Encoder Loss Ablations | | | | | |
| CNN | DistilBERT (p) | Cosine distance | FiLM (demo), Concat (lang) | DeL-TaCo (ours) | 12.2 ± 2.5 |
| CNN | DistilBERT (p) | None | FiLM (demo), Concat (lang) | DeL-TaCo (ours) | 12.5 ± 2.6 |
| Task Conditioning Architecture Ablations | | | | | |
| CNN | DistilBERT (p) | Contrastive | Concat (demo + lang) | DeL-TaCo (ours) | 10.6 ± 4.2 |
| CNN | DistilBERT (p) | Contrastive | FiLM (demo + lang) | DeL-TaCo (ours) | 17.3 ± 2.7 |

## F  ABLATION ANALYSIS

In Table 8, we compare the performance of different language encoders, demonstration encoders, task encoder loss types, and task conditioning architectures in Experimental Scenario A to our best model from Table 1 (top row).

**Language Encoder Ablations.** Finetuning the language model did not improve performance, most likely because the model becomes significantly harder to train jointly with a relatively deep language encoder. Perhaps training the policy and finetuning the language model in separate stages may yield better results. We also compare with other language models such as DistilRoBERTa (Sanh et al., 2020) and miniLM (Wang et al., 2020). These were chosen because they have (1) a relatively small number of parameters for computational efficiency, and (2) were shown to achieve high performance on language-conditioned robotic policies when compared to other common language models (Mees et al., 2022). When using CLIP as the demo and language encoder, we did not notice much of a performance difference from adding a finetuneable MLP head on top of the frozen CLIP language encoder.

**Demonstration Encoder Ablations.** Using pretrained demonstration encoders, such as R3M, did not perform better than training a small CNN from scratch—a similar takeaway from our experiments with pretrained CLIP.

**Loss Ablations.** Our approach outperforms using a BC-Z-styled cosine-distance loss term $\mathcal{L}_{demo}(z_{demo}, z_{lang}) = 1 - z_{demo} \cdot z_{lang}$ (where $\|z_{demo}\| = \|z_{lang}\| = 1$). Our contrastive task encoder loss term is crucial; without it (setting $\alpha = 0$ in Eqn. 3), performance drops substantially.

**Task Conditioning Architecture Ablations.** Concatenating $z_{demo}$ and $z_{lang}$ before feeding into FiLM layers causes a slight drop in performance compared to our architecture in Figure 2, and simply concatenating $z_{demo}$ and $z_{lang}$ to the image observation embeddings without FiLM dramatically decreases performance.

## G  ADDITIONAL AMBIGUITY EXPERIMENTS

In Tables 1 and 2, we examined the effects of conditioning the multitask policy on a demonstration that is ambiguous about the target container, plus an unambiguous language instruction that clearly specifies both the target object and container for each pick-and-place task. We label this as ambiguity scheme (i).

To further examine the utility of multi-modal task conditioning, we experimented with a different ambiguity scheme (ii) in which the language instruction is *ambiguous* about the target *object* but *unambiguous* about the target *container*, and the demonstration is *unambiguous* about the target *object* but *ambiguous* about the target *container*. This scheme allows us to test how well two task-conditioning modalities complement each other if each modality unambiguously conveys only a single aspect of the task.

Table 9: Ambiguity Experiments

| Ambiguity Scheme | Demo Ambiguity | Language Ambiguity | Task Conditioning | Success Rate $\pm$ SD (%) |
|---|---|---|---|---|
| | – | None | Language-only | $10.4 \pm 1.6$ |
| (i) (From Table 1) | container | – | Demo-only | $14.6 \pm 2.2$ |
| | container | None | DeL-TaCo (ours) | $\mathbf{19.9 \pm 1.8}$ |
| | – | object | Language-only | $15.8 \pm 2.3$ |
| (ii) | container | – | Demo-only | $22.7 \pm 2.4$ |
| | container | object | DeL-TaCo (ours) | $\mathbf{26.7 \pm 5.2}$ |

To introduce ambiguity on the target objects, in scheme (ii), we place two identical objects on the scene in different parts of the tray, plus a third visually distinct distractor object. Exactly one of these two identical objects is the target object. The language instruction is ambiguous because it does not convey the positional identifier that describes which side of the tray the target object is at. However, the demonstration unambiguously conveys the target object by showing the robot picking up the target object from the correct part of the tray.

The results are shown in Table 9. By displaying two identical objects as we do in scheme (ii), we visually reveal that the target object is one of the two identical objects (and not the third, distinct object on the scene), causing scheme (ii) to have higher overall success rates than scheme (i). We see that the gap between DeL-TaCo and the demo-only policy slightly decreases from more than $5\%$ in scheme (i) to $4\%$ in scheme (ii) because the language instruction unambiguously specifies the task in scheme (i), complementing the demonstration which is ambiguous on both the target object and container, whereas in scheme (ii), both the language instruction and demonstration are ambiguous in one aspect (either in specifying the container or the object), narrowing the performance gap between DeL-TaCo and the demo-only policies.

## H   DEMONSTRATION FORMATTING FOR $f_{demo}$

We represent each demonstration as an $m \times n$ image array consisting of observations from the first timestep, the last timestep, and $mn - 2$ other randomly selected timesteps from the trajectory, arranged in raster-scan order. For our CLIP and R3M experiments, we use $(m, n) = (2, 2)$ because CLIP and R3M perform a center square crop on each input image, so we made the demonstration array square. When using our trained-from-scratch $f_{demo}$, we used $(m, n) = (1, 2)$ for computational efficiency. This sufficed for our tasks because pick-and-place was not a particularly long horizon task, so including more frames did not improve performance.

