# OpenReview forum: "Using Both Demonstrations and Language Instructions to Efficiently Learn Robotic Tasks"
_ICLR.cc/2023/Conference — ICLR 2023 poster_

### Official Review · Reviewer_5Uyd · 2022-10-21

**Confidence:** 5
**Correctness:** 4
**Technical Novelty And Significance:** 2
**Empirical Novelty And Significance:** 2
**Recommendation:** 6

**Clarity, Quality, Novelty And Reproducibility:**

Clarity:
* The work is clearly explained and the method is easy to understand.
* The task description is more difficult to understand directly from the text, but the appendix helps to understand what objects were used, what "shape" means and how it is distinct from "object", etc.

Quality:
* The work is high quality and appears to have been conducted rigorously, though natural questions around training details or specifications appear to be left unanswered.

Novelty:
* The work has not compared thoroughly to related work, except through ablations of the proposed method.
* Prior work on language encoding for goal-conditioned agents is missing [1, 2]
* The overall method is a relatively straightforward combination of prior work without significant additional contributions, though this is not a reason to reject as the method is well-explained, well-motivated, and performs well.

[1] Sodhani, Shagun, Amy Zhang, and Joelle Pineau. "Multi-task reinforcement learning with context-based representations." International Conference on Machine Learning. PMLR, 2021.

[2] Silva, Andrew, et al. "LanCon-Learn: Learning With Language to Enable Generalization in Multi-Task Manipulation." IEEE Robotics and Automation Letters 7.2 (2021): 1635-1642.

**Strength And Weaknesses:**

Strengths:
* The method is clearly described and intuitive, and is a natural next-step after prior work.
* Combination of demonstrations and language appears to provide significant improvements over either modality individually.
* The work is positioned clearly with respect to BC-Z.
* The authors developed a simulator/simulated task for their work.

Weaknesses:
* The proposed method feels like a solid first-step, but many natural follow-up questions are left unexplored (what if the language encoder is fine-tuned? Why not pass the CLIP embeddings through a small, fine-tuned MLP? How does a pre-trained vision module work for the demo encoder? etc)
* Training the demonstration encoder seems to require prior knowledge on the number of possible tasks, limiting the generality of DeL-TaCo  to situations with fixed numbers of tasks, despite the goal-encoding being quite general.
* Experimental reporting feels disjointed and incomplete. While the discussion seems to suggest that language is superior to demonstrations (e.g., saying that language is worth 50 demonstrations), in fact the demo-only method performs better than the language-only method in both experiments. There are also no upper-bound numbers for Table 2, so we can see that DeL-TaCo is better than demo/language-only, but we don't know how far it is from "perfect" performance.

**Summary Of The Paper:**

The paper proposes Joint Demo-Language Task Conditioning, DeL-TaCo, as a new way of training and specifying goals for goal-conditioned task-learning agents. DeL-TaCo learns a demonstration encoder and uses a frozen, pre-trained language encoder to embed task demonstrations and language-specifications for simulated robots to perform pick-and-place tasks with various objects/colors/shapes. DeL-TaCo is compared to language-only and demonstration-only ablations, showing that both demonstrations and language are important to achieve the highest success rates, and two different encoders are compared, a pre-trained CLIP vs. a pre-trained DistilBERT + trained demonstration encoder.

**Summary Of The Review:**

The proposed method, DeL-TaCo, is explained clearly, is intuitive and straightforward, and appears to work well. The paper suffers from incomplete reporting in the results, a relative lack of novelty, and unanswered questions about training details or design decisions. With simple additional experiments around fine-tuned language encoders, clearer results for demonstration-vs-language goal-specification and performance upper-bounds, and position to recent related works, I would advocate for acceptance. As the paper reads currently, it does not seem publication-ready.

---

> ### Author Response · Authors · 2022-11-19
> **Response to Reviewer 5Uyd**
>
> Thank you for your feedback and questions. In experimentally investigating your questions, we gained additional insights and will summarize them in response to each of your questions/comments below.
>
> > Many natural follow-up questions are left unexplored (what if the language encoder is fine-tuned? Why not pass the CLIP embeddings through a small, fine-tuned MLP? How does a pre-trained vision module work for the demo encoder?)
>
> We ran experiments to answer all three of these questions (see [Overall Response Section 1](https://openreview.net/forum?id=4u42KCQxCn8&noteId=5gUoRqWlxVX) for detailed results).
>
> - Finetuning the language encoder dramatically slows down training and does not improve DeL-TaCo performance (see Overall Response Section 1.A.i).
> - Having a finetuned MLP head on top of the frozen CLIP language encoder does not seem to meaningfully affect performance (see Overall Response Section 1.A.iii).
> - We did experiment with a pre-trained demo encoder (CLIP) in the original version of the paper. In our updates, we run additional experiments with pre-trained R3M as the demo encoder, which also did not perform better than training the demo encoder from scratch (see Overall Response Section 1.B).
>
> > Training the demonstration encoder seems to require prior knowledge on the number of possible tasks, limiting the generality of DeL-TaCo to situations with fixed numbers of tasks, despite the goal-encoding being quite general.
>
> Training the demo encoder does *not* require prior knowledge on the number of possible tasks. The demo encoder is not explicitly given knowledge of how many training or test tasks there are; we only provide (demo, language) pairs for different tasks as training data. Perhaps the misunderstanding here arises from how we calculate the contrastive learning loss. We have a task batch size for contrastive learning (this is the variable $k$ in section 4.2; we set $k=16$ in our experiments), so 16 random (demo, language) pairs from different tasks are sampled to compute the contrastive loss, and the Identity Matrix in Equation 1 is 16x16. Thus, our approach is not limited to situations with fixed numbers of tasks. Finetuning the policy on additional tasks shouldn’t be a problem, though we did not empirically try this.
>
> > Experimental reporting feels disjointed and incomplete. While the discussion seems to suggest that language is superior to demonstrations (e.g., saying that language is worth 50 demonstrations), in fact the demo-only method performs better than the language-only method in both experiments.
>
> By “language is worth 50 demonstrations,” we didn’t mean that language is “superior” to demonstrations. We meant to say that providing a language instruction in addition to a demonstration, as done in the DeL-TaCo policy, achieves the same performance as fine-tuning the demo-only policy on 50 demonstrations per test task. In essence, learning from the combination of demonstrations and language performs better than learning from only demonstrations. This finding is not incompatible with the fact you bring up--that the demo-only method performs better than the language-only method in both Exp. Scenarios A and B.
>
> > There are also no upper-bound numbers for Table 2, so we can see that DeL-TaCo is better than demo/language-only, but we don't know how far it is from "perfect" performance.
>
> We have added upper and lower-bound numbers to Table 2 and reran our upperbounds for Table 1 in the updated paper. Also see [Overall Response Section 4](https://openreview.net/forum?id=4u42KCQxCn8&noteId=UD5GxSQtKb).
>
> > The work has not compared thoroughly to related work, except through ablations of the proposed method.
>
> We have added comparisons to MCIL and BC-Z, two of the most relevant recent work learning from demonstrations and/or language. DeL-TaCo achieves roughly double the success rate of both of these prior works. See [Overall Response Section 2](https://openreview.net/forum?id=4u42KCQxCn8&noteId=UD5GxSQtKb).
>
>
> > Prior work on language encoding for goal-conditioned agents is missing [1, 2].
>
> These have been added to our related work section in the updated paper.
>
> > With simple additional experiments around fine-tuned language encoders, clearer results for demonstration-vs-language goal-specification and performance upper-bounds, and position to recent related works, I would advocate for acceptance.
>
> We believe our rebuttal and updated paper meets the criteria you spelled out for a potential acceptance: fine-tuned language encoder experiments, Exp. Scenario B upper and lower bounds, comparison to prior works BC-Z and MCIL (and DeL-TaCo outperforms both), additional ablations, and clarifications for the “worth of language” experiments.
>
> ### Concluding Remarks
> If there are other aspects of the results section that we can make clearer, or any other experimental results you would like to see, please let us know. We would be glad to answer any further questions you may have. Thank you.

---

> ### Author Response · Authors · 2022-12-10
> **Brief Check-in Before End of Phase 2 Discussion Period**
>
> Just wanted to check in with you to see if you had any questions/concerns/feedback after reading our rebuttal, since discussion phase 2 ends on 12/12, which is the last chance for us to directly respond to any additional comments you may have. Thank you for your time in advance!

---

### Official Review · Reviewer_Ch3U · 2022-10-23

**Confidence:** 4
**Correctness:** 3
**Technical Novelty And Significance:** 3
**Empirical Novelty And Significance:** 3
**Recommendation:** 6

**Clarity, Quality, Novelty And Reproducibility:**

**Writing**

6. How were the demonstrations collected? : On page 7 under the sub-section of *Data*, the authors mention that a "scripted policy" was used to collect demonstrations. What was this scripted policy -- a policy that uses inverse kinematics? Please mention the details of it.

**Section 4.1**

7. The variables $m$ and $n$ are introduced in the first paragraph and nowhere in the main paper are they defined. It is only in the appendix that their descriptions are given. Please define what they are for clarity.

8. Under the *Task Conditioning Architecture* sub-section, please briefly describe what *FiLM* does. One sentence should be sufficient -- otherwise, it makes the reader switch back and forth between the original paper and this.


**Visual Aesthetics (not considered for decision-making)**

9. I believe adding plots instead of the table would be much easier to read (except for Table 3). I say this after looking that the authors have used plots on the project webpage, which have better readability and also look better.

**Reproducibility**

10. The authors do mention the details of the model in the appendix including the hyperparameters they used. However, I would like to ask if the authors plan to open-source their code, if accepted to the conference?

**Strength And Weaknesses:**

**Strengths**

1. The paper is mostly well-written, and the methodology section is easy to understand in one go.

2. Section 5.3 which compared how much was a language worth was a good discussion -- which drives the point that having language as input is indeed beneficial (as corroborated by several other works in the field)!


**Weaknesses + Questions:**

**Claims**
1. The authors claim the following in Sec 2.2

> "We argue that conditioning the policy on both a demonstration and language not only ameliorates the ambiguity issues with language-only and demonstration-only specifications but is much easier and more cost-effective for the end-user to provide."

I'm not sure what the authors mean by "more cost-effective for the end-user to provide" -- I agree that providing numerous demonstrations (at least 50 according to the experiments) has a high cost on the user, but having only language would have a lower cost (as done in [1])!


**Methodology**

2. The rationale behind the Task Conditioning Architecture: How was the architecture selected? Was it based on previous works of [2], and [3]?

    A simple variant (baseline) would be to concatenate the $z_{demo}$ and $z_{lang}$ and use that as input to the policy. Was this baseline tried? If so, what were the observations?

**Experiments**

3. Success Rate: I am not sure why this complicated way of computing the Success rate was adopted. Specifically, I'd like the authors to address the following questions:

   a) Why are the rollouts *during* the training being considered? What is the problem with computing the success rate after training (800-900k steps)?

   b) How does the current definition of success rate avoid *cherry-picking* when compared to (a)?

4. Training + Test Sets (a Scenario C): In all the experiments, the three objects in the scene are *visually distinct*. Having the same objects with a) the same colors, as well as b) different colors in the test set would be a crucial experiment to test the generalization of the method. I would like to see the results of such an experiment. Since it is only testing, I believe it will not take a lot of time.

5. Baseline Comparison: I would expect a comparison with [1] to be present in the paper. I understand that the code for [1] is unfortunately not open-sourced (which I don't hold against you, and blame the authors of [1]), but the method in [1] can be adapted to your codebase wherein you have a shared latent space -- which is essentially the crux of [1].


-------
**References**

[1] Corey Lynch and Pierre Sermanet. Language conditioned imitation learning over unstructured data. In RSS 2021.

[2] Eric Jang, Alex Irpan, Mohi Khansari, Daniel Kappler, Frederik Ebert, Corey Lynch, Sergey Levine, and Chelsea Finn. BC-z: Zero-shot task generalization with robotic imitation learning. In CoRL 2021.

[3] Ethan Perez, Florian Strub, Harm de Vries, Vincent Dumoulin, and Aaron C. Courville. Film: Visual reasoning with a general conditioning layer. In AAAI, 2018.

**Summary Of The Paper:**

The paper argues that solely using language or video demonstrations to learn robotic manipulation tasks is inadequate as it creates ambiguities in learning. To alleviate this confusion, the authors propose to train a multi-task policy that condition on *both* the video demonstrations as well as the language.

They work in the paradigm of imitation learning, wherein they train a single policy for all the robotic pick-and-place tasks.

**Summary Of The Review:**

Even though the paper's contribution in methodology is significant enough for the community, due to the lack of extensive experiments and proper baselines as mentioned in the Weaknesses section, I would vote 5 : *marginally below the acceptance threshold*. However, my final decision is subject to change based on the authors' rebuttal and comments of other reviewers.

---

> ### Author Response · Authors · 2022-11-05
> **Clarification Question on Item 4**
>
> Thank you very much for your review. As we prepare our responses, we would like to ask you a clarifying question about your fourth point on “Training + Test Sets (a Scenario C).”
>
> We’re a bit confused by your comment about testing on the same objects with “a) the same colors, as well as b) different colors.” Did you mean that for each of our 24 training objects $o$, each of which appear in a certain color $color_{train}(o)$ during training, we should evaluate on a new Scenario C with each training object appearing during testing in a different color $color_{test}(o)$, where $color_{test}(o) \ne color_{train}(o), \forall o \in$ training object set?

---

> > ### Comment · Reviewer_Ch3U · 2022-11-05
> > **RE: Clarification Question on Item 4**
> >
> > Thank you for asking for a clarification. I should have probably expanded on this point a bit more in my official review.
> >
> > I believe the description that you mention ($color_{test}(o) \neq color_{train}(o)$) is being done in Scenario B -- Novel Colors and Shapes.
> >
> > My comment was regarding the objects being *distinct* in all the scenarios (A & B). Concretely, I'd like you to:
> >
> > a) *test* the model on settings which have *at least* two same objects of the same color. The rationale here is that language should provide the grounding to the agent so as to which object to manipulate (since they are identical in all aspects except their position and pose).
> >
> > b) *test* the model on settings which have *at least* two same objects of the *different* color. Here the grounding should be on color, which my hypothesis is that DeL-TaCo should be able to handle well.
> >
> > Let me know if this description was clear and if any of the experiments : (a) or (b) have already been reported the paper, please let me know as well -- as far as I know these have not been performed.
> >
> > Thanks again for reaching out early in the process.

---

> > > ### Author Response · Authors · 2022-11-06
> > > **Example Instantiation of Item 4 evaluations**
> > >
> > > Thanks for your prompt response. We’d like to verify we understood your proposed (a) and (b), and that our plan to show these results is aligned with your response. We plan to take the policy checkpoints trained from scenario B and test DeL-TaCo, demo-only, and language-only policies on (a) and (b). We wrote an example situation for each based on our interpretation of your description.
> > >
> > > Example of (a):
> > > - **Scene objects ($o_1, o_2, o_3$) during evaluation:** $o_1$ in one of the two left quadrants of the workspace, $o_2$ in one of the two right quadrants of the workspace, and $o_3$ in any quadrant. $o_1$ and $o_2$ are visually identical (same object, same color, potentially different pose). Let’s say $o_1 = o_2 =$ “red bottle” and $o_3 =$ “gray glass.”
> > > - **Language instruction:** “Put left red bottle in front container.”
> > > - **Demonstration:** The objects on the scene in the demonstration are arranged similarly to what the robot sees during evaluation in that $o_1$ is on the left, $o_2$ is on the right, and $o_3$ can be on either side, $o_1 = o_2 =$ “red bottle,” and $o_3$ is a distractor object (not necessarily a “gray glass”). The demonstration shows the robot putting the left bottle ($o_1$) into the front container.
> > >
> > > Example of (b):
> > > - **Scene objects during evaluation:** Similar to situation (a)’s evaluation scene, but now $o_1$, $o_2$, and $o_3$ may be in any quadrant of the workspace, $o_1 =$ “red bottle,” $o_2 =$ “blue bottle,” and $o_3$ is some other non-bottle object neither red nor blue.
> > > - **Language instruction:** “Put blue colored object in green container.”
> > > - **Demonstration:** The objects on the scene in the demonstration consist of a red bottle, blue bottle, and a third distractor likely not the same as $o_3$ in the evaluation scene (but is still a non-bottle, non-red, non-blue object). Again, all three objects may be in any quadrant of the workspace. The demonstration shows the robot putting the blue bottle into the green container.
> > >
> > > As you probably suspect, (a) seems challenging in part because directional descriptors for target objects were never seen during training (though the containers were sometimes directionally described with words like “left/right/front/back”).
> > >
> > > Let us know if this is a valid instantiation of (a) and (b) you had in mind. If so, we will try to get aggregated results across different objects (not just bottles) and target container identifiers (not just the “front” or “green” container).
> > >
> > > =====
> > >
> > > Brief side note: our scenario B does not actually do what we described above in terms of $color_{test}(o) \ne color_{train}(o)$ because each of the 32 train and test objects has its own fixed color (shown in figure 5, Appendix A.3). This means that the same object does not appear in different colors between training and testing. In the novel colors situation, we have color tasks of the instruction form “put [color] colored object in [container identifier] container,” where the color is one of {white, red, orange, yellow} during training and {black and white, brown, blue, gray} during testing. This can be confusing, and we’d be happy to explain this in further detail, though this is not the main purpose of the thread.

---

> > > > ### Comment · Reviewer_Ch3U · 2022-11-07
> > > > **RE: Example Instantiation of Item 4 evaluations**
> > > >
> > > > Thanks for giving examples for (a) and (b).
> > > > Yes examples that you have given for (a) and (b) are indeed similar to ones that I've on my mind.
> > > >
> > > > Thanks for the side note on scenario B.

---

> > > > > ### Comment · Reviewer_Ch3U · 2022-11-17
> > > > > **Just checking in for the rebuttal**
> > > > >
> > > > > Hi authors,
> > > > >
> > > > > We are nearing the rebuttal deadline, so please let me know if you have any further clarification questions. And I would be expecting the rebuttal responses sometime soon.
> > > > >
> > > > > Thank you!

---

> > > > > > ### Author Response · Authors · 2022-11-17
> > > > > > **Re: Just checking in for the rebuttal**
> > > > > >
> > > > > > Yes, thanks for the reminder! We have been implementing and running the requested experiments/comparisons and will post the rebuttal responses and updated paper by tomorrow (Nov 18) AoE.

---

> ### Author Response · Authors · 2022-11-19
> **Response to Reviewer Ch3U (Part 2/2)**
>
> > 4. Training + Test Sets (a Scenario C): In all the experiments, the three objects in the scene are visually distinct. Having the same objects with a) the same colors, as well as b) different colors in the test set would be a crucial experiment to test the generalization of the method. I would like to see the results of such an experiment.
>
> Thank you for your earlier clarifications on this topic. We implemented two environments, one for case (a) and one for case (b). We defined 128 tasks for Case (a) and 192 tasks for Case (b). The success rates below are computed by taking the latest policy checkpoint of 3 seeds for each task conditioning scheme (trained under Exp. Scenario A) and running a total of 768 trials for Case (a) and 1152 trials for Case (b). (We did not have earlier checkpoints saved, preventing us from computing these success rates with the method we defended earlier.)
>
> Case (a):
> - language-only: 4.9%
> - demo-only: 9.6%
> - DeL-TaCo: 8.7%
>
> Case (b):
> - language-only: 12.0%
> - demo-only: 10.7%
> - DeL-TaCo: 14.0%
>
> Our policies perform relatively poorly in case (a) most likely because the language instructions refer to objects by their spatial locations, which was not seen before during training. This results in language embeddings largely incoherent to the policy, slightly hurting the performance of DeL-TaCo relative to demo-only. The success rates of Case (b) are also lower than those in Tables 1 and 2 perhaps because we changed the color and texture of *every object* to be different from its appearance during training, potentially throwing off the internal object name-to-visual feature mappings learned by the policies.
>
> This was an interesting evaluation to implement and run--thanks for the suggestion and earlier guidance.
>
> > 5. Baseline Comparison: I would expect a comparison with [1] to be present in the paper.
>
> This is now shown in the updated paper. Please see [Overall Response Section 2](https://openreview.net/forum?id=4u42KCQxCn8&noteId=UD5GxSQtKb) for this comparison of DeL-TaCo with MCIL. MCIL attains less than half the success rate of DeL-TaCo in our problem setting.
>
> > 6. How were the demonstrations collected? On page 7 under the sub-section of Data, the authors mention that a "scripted policy" was used to collect demonstrations. What was this scripted policy -- a policy that uses inverse kinematics? Please mention the details of it.
>
> Please see Appendix C for details of the scripted policy. At a high level, the scripted policy simply computes the delta xyz action based on the current end effector position and the target object/target container position. The scripted policy does not directly perform IK because we work in xyz action space with xyz controllers.
>
> > 7. The variables m and n are introduced in the first paragraph and nowhere in the main paper are they defined. It is only in the appendix that their descriptions are given. Please define what they are for clarity.
>
> Added the integer values of $m,n$ in the first paragraph of Section 4 in the updated paper.
>
> > 8. Under the Task Conditioning Architecture sub-section, please briefly describe what FiLM does. One sentence should be sufficient -- otherwise, it makes the reader switch back and forth between the original paper and this.
>
> We have provided a brief description of FiLM in that paragraph in the updated paper. Perhaps Figure 7 (“Detailed architecture”) may also help in understanding FiLM and how it fits in each residual block of the ResNet.
>
> > 9. I believe adding plots instead of the table would be much easier to read (except for Table 3). I say this after looking that the authors have used plots on the project webpage, which have better readability and also look better.
>
> We have added plots corresponding to Tables 1-3 in Appendix E.
>
> > 10. The authors do mention the details of the model in the appendix including the hyperparameters they used. However, I would like to ask if the authors plan to open-source their code, if accepted to the conference?
>
> If accepted, we intend to spend at least a week to clean up, organize, and open-source our code as we believe this is a very important part of the scientific process. We will also try to respond to github issues and questions in a timely manner whenever they are posted.
>
> ### Concluding Remarks
>
> Thanks again for your feedback, suggestions, and well-organized review. We hope we’ve answered all of the issues and questions you raised. Please let us know if you have additional questions or suggestions.

---

> ### Author Response · Authors · 2022-11-19
> **Response to Reviewer Ch3U (Part 1/2)**
>
> Thank you very much for your detailed review and for your clarifications in our earlier exchange. We respond to your suggestions and questions item-by-item below.
>
> > 1. I'm not sure what the authors mean by ‘more cost-effective for the end-user to provide’ -- I agree that providing numerous demonstrations (at least 50 according to the experiments) has a high cost on the user, but having only language would have a lower cost (as done in [1])!
>
> We’re simply arguing that *to achieve the same level of performance*, providing a single demonstration and language instruction to DeL-TaCo is more cost-effective than providing 50 demonstrations of the task to a demo-only policy, given that both of them have the same performance in our problem setting. Specifically in that paragraph, we are trying to argue that certain tasks are complex enough where language-only specification would require very fine-grained, detailed, long instructions, which can be a lot harder to provide than a brief, relatively high-level instruction paired with a single demo.
>
> > 2. The rationale behind the Task Conditioning Architecture: How was the architecture selected? Was it based on previous works of [2], and [3]? A simple variant (baseline) would be to concatenate the z_lang and z_demo and use that as input to the policy. Was this baseline tried? If so, what were the observations?
>
> We use the architecture of feeding $z_{demo}$ through the policy FiLM layers and concatenating $z_{lang}$ to the image observation embeddings. BC-Z (which you labeled as [2]) uses FiLM (which you labeled as [3]), which is what we based our architecture on. See [Overall Response Section 1.D](https://openreview.net/forum?id=4u42KCQxCn8&noteId=5gUoRqWlxVX) for the ablations we ran, including the simple baseline you mention of concatenating $z_{lang}$ and $z_{demo}$. In short, our architecture performs a few points better than what you asked about, and a lot better than not using FiLM at all and just concatenating the task embeddings to the image observation embeddings.
>
> We came to our final architecture after a lot of experimentation. For instance, we found that if the demo-only policy performs better than the language-only policy, it is better to feed $z_{demo}$ through FiLM and concat $z_{lang}$ to the image observation embeddings when training DeL-TaCo. In other words, the more “important” embedding should be passed through FiLM while the less important embedding should be concatenated to the image observation embeddings. This makes sense since networks pay more attention to embeddings passed through FiLM than embeddings concatenated to an intermediate representation in the final few layers.
>
> > 3. Success Rate: I am not sure why this complicated way of computing the Success rate was adopted. (a) Why are the rollouts during the training being considered? What is the problem with computing the success rate after training (800-900k steps)? (b) How does the current definition of success rate avoid cherry-picking when compared to (a)?
>
> (a) The problem with selecting the last checkpoint from training is that the average success rate (even with hundreds of rollouts in an evaluation epoch) is quite variable during the course of training on a single seed, making metrics such as comparing the last checkpoint or the best checkpoint prone to high variability. Sometimes the evaluation performance fluctuates by 5% in consecutive training epochs. Hence we take the top 10 evaluation epochs during training, ranked by average success rate across the three seeds, and average those to get a large sample size when calculating the success rate. Since performance generally improves over time, most of the top 10 best performing epochs are from the last 200k steps of training.
>
> (b) Given the high variability of success rates from epoch to epoch, we thought it would be misleading (and too prone to noise) to simply compare a single checkpoint (such as the best or latest checkpoint) between the methods. Thus we use the system described earlier (and in Appendix D) to be more robust to epoch-to-epoch success rate variability.

---

> ### Author Response · Authors · 2022-12-10
> **Brief Check-in Before End of Phase 2 Discussion Period**
>
> Just wanted to check in with you to see if you had any questions/concerns/feedback after reading our rebuttal, since discussion phase 2 ends on 12/12, which is the last chance for us to directly respond to any additional comments you may have. Thank you for your time in advance!

---

> > ### Comment · Reviewer_Ch3U · 2022-12-13
> > **Final decision**
> >
> > I thank the authors for updating the manuscript and doing all the experiments that I and other reviewers have asked for.
> > I'm very much satisfied with the Case (a) and Case (b) [Scenario C] results. This proves that there is a scope to improve the generalization ability of the method (which is something that the authors can definitely consider in the future).
> >
> > Also thanks for the explanation for the success rates calculation.
> >
> > To reflect this, I have updated my review to 6 (marginally above the acceptance threshold).

---

> > > ### Author Response · Authors · 2022-12-13
> > > **Re: Final decision**
> > >
> > > Thank you very much for taking the time to read our rebuttal + the updated manuscript, and thank you for reconsidering!

---

### Official Review · Reviewer_p2aG · 2022-10-25

**Confidence:** 3
**Correctness:** 4
**Technical Novelty And Significance:** 2
**Empirical Novelty And Significance:** 2
**Recommendation:** 6

**Clarity, Quality, Novelty And Reproducibility:**

The paper is relatively clear, and the figures are quite nice.

The novelty of the paper is relatively minor. Prior work has explored conditioning on language and demonstrations, though, the combination of the two is more unique.

The paper does not appear to have code for reproducibility, though, additional details are provided in the appendix.

**Details Of Ethics Concerns:**

This paper builds upon large language models, and the ethical issues of those are not discussed in the limitations. This can readily be corrected by the authors.

**Strength And Weaknesses:**

Strengths:

The paper clearly describes a supervised learning approach to multi-task learning with tasks conditioned on both a natural language instruction and a demonstration of the task.

The figures are relatively informative.

Pseudocode is provided.

The paper reports positive, empirical results on a relatively large dataset of multiple tasks.

Table 3 shows how much data is required to enable a demo-only policy to match the performance of the proposed method, DeL-TaCo, showing the proposed method is better up until reaching ~50 demonstrations for the dataset for the novel task.

A limitations and future work section is included.

Weaknesses:

This approach requires both language and instructions to function. As such, one would want to show superior performance to methods that require only one. For example, recent work by Wen et al. (2022) required only a single demonstration on a novel task:

Wen, B., Lian, W., Bekris, K. and Schaal, S., 2022. You Only Demonstrate Once: Category-Level Manipulation from Single Visual Demonstration. arXiv preprint arXiv:2201.12716.

On the language side, there are quite a few recent papers show strong performance conditioning on only language:

Sodhani, S., Zhang, A. and Pineau, J., 2021, July. Multi-task reinforcement learning with context-based representations. In International Conference on Machine Learning (pp. 9767-9779). PMLR.

Silva, A., Moorman, N., Silva, W., Zaidi, Z., Gopalan, N. and Gombolay, M., 2021. LanCon-Learn: Learning With Language to Enable Generalization in Multi-Task Manipulation. IEEE Robotics and Automation Letters, 7(2), pp.1635-1642.

Nair, S., Mitchell, E., Chen, K., Savarese, S. and Finn, C., 2022, January. Learning language-conditioned robot behavior from offline data and crowd-sourced annotation. In Conference on Robot Learning (pp. 1303-1315). PMLR.

Shridhar, M., Manuelli, L. and Fox, D., 2022, January. Cliport: What and where pathways for robotic manipulation. In Conference on Robot Learning (pp. 894-906). PMLR.

This reviewer notes that the authors discussed CLIPort and the Nair et al. (2022) paper; however, these works were not benchmarked against. The paper does not present a clear argument against benchmarking against these approaches (other than the implicit argument that they only condition on language, which is actually an advantageous quality).

This paper is a relatively straightforward design of a neural network-based architecture where embeddings are learned for language instructions and demos (which are forced to be the same through a contrastive loss) and then fed into a policy head for supervised learning. It is unclear whether new theories or insights are gleaned (other than adding more input data can enhance performance by 5-9%).

More insight into why DistilBert was used could be helpful. What would other language embeddings look like?

An ablation study showing the efficacy of the contrastive loss and other components would be helpful.


**Summary Of The Paper:**

This paper develops an approach, Joint Demo-Language Task Conditioning (DeL-TaCo), for behavior cloning in a multi-task setting where the task is specified by both language and a demonstration. The approach includes a contrastive loss to guide the learning of the language and demonstration embeddings, which are then fed into a policy head. Results are conducted in simulation and show that virtual robot manipulation tasks can be learned by DeL-TaCo and that this approach generalizes better to novel tasks.

**Summary Of The Review:**

This paper provides positive results for conditioning on language and demonstrations in multi-task learning for zero-shot generalization. However, the paper would be improved by more baselines and a more compelling case that the results are novel and significant.

---

> ### Author Response · Authors · 2022-11-19
> **Response to Reviewer p2aG (Part 2/2)**
>
> > More insight into why DistilBert was used could be helpful. What would other language embeddings look like?
>
> Please see [Overall Response Section 1.A.ii](https://openreview.net/forum?id=4u42KCQxCn8&noteId=5gUoRqWlxVX) (and Appendix F), where we compare to not only the CLIP language encoder, but also DistilRoBERTa and miniLM. Regardless of the language encoder, DeL-TaCo consistently outperforms the corresponding language-only policy.
>
> > An ablation study showing the efficacy of the contrastive loss and other components would be helpful.
>
> Please see [Overall Response Section 1.C](https://openreview.net/forum?id=4u42KCQxCn8&noteId=5gUoRqWlxVX) (and Appendix F), where we specifically compare to (i) cosine distance loss and (ii) not having any task encoder loss term at all. Both of these approaches perform considerably worse than DeL-TaCo, which uses a contrastive task encoder loss term.
>
> > This paper builds upon large language models, and the ethical issues of those are not discussed in the limitations.
>
> We updated the paper with an ethics statement (Section 8) that addresses some of the concerns from using LLMs.
>
> ### Concluding remarks.
>
> We understand why you might feel skeptical about the utility of demo + language task conditioning as opposed to policies that are solely language-conditioned. However, even with current state-of-the-art approaches, language-conditioned policies face complex grounding challenges in new environments. Additionally, only having language as a modality for learning/specifying new tasks can be quite challenging, such as in tasks with temporally-extended motions that are hard to describe with language alone. Providing a single demonstration along with the language instruction provides valuable information for grounding in performing new tasks.
>
> We do seem to agree on the fact that DeL-TaCo does something that prior work does not, in terms of learning new robotic tasks during training and testing with both demonstrations and language. We hope you get a chance to read our extended motivation argument in [Overall Response Section 3](https://openreview.net/forum?id=4u42KCQxCn8&noteId=UD5GxSQtKb), which elaborates on why we believe it is important to be able to learn from both demonstrations and language, and why single-modality task specification, as previous works have done, seems to unnecessarily bottleneck generalization performance.
>
> We've thought deeply about your feedback and have used it to extend our ablations and more thoroughly reason about the importance of using demos and language in robot learning. We hope you will consider some of the arguments in this rebuttal and the improvements made in our paper update. Feel free to ask any additional questions you may have or express any additional concerns about our method, and we'll do our best to answer them. Thank you.

---

> > ### Comment · Reviewer_p2aG · 2022-11-20
> > **Response**
> >
> > I appreciate the response from the authors and will improve my score.

---

> > > ### Author Response · Authors · 2022-11-20
> > > **Re: Response**
> > >
> > > Thank you very much for reading our response and for reconsidering!

---

> ### Author Response · Authors · 2022-11-19
> **Response to Reviewer p2aG (Part 1/2)**
>
> Thank you very much for your review. You raise a number of good points, and we address them below.
>
> > This approach requires both language and instructions [most likely meant “demonstrations”] to function. As such, one would want to show superior performance to methods that require only one.
>
> Our method uses multimodal task embeddings. We compare to the state-of-the-art approaches in this field that are most relevant to ours: BC-Z [1] and MCIL [2], both of which require only one modality during evaluation. DeL-TaCo achieves more than double the success rate of both BC-Z and MCIL (see [Overall Response Section 2](https://openreview.net/forum?id=4u42KCQxCn8&noteId=UD5GxSQtKb)).
>
> During the rebuttal period, we did not get a chance to additionally compare to some of the language-only approaches you listed, but we have added the missing language-only papers (Sodhani, et. al. and Silva, et. al.) to our related work section. However, we did not find the demonstration-only paper you brought up (Wen, et. al.) to be naturally comparable to our method, since it takes a keypoint-centric object representation approach that requires path-planning, projecting demonstrations into the environment, and perception from depth sensors, neither of which is applicable in our approach, since our policy performs end-to-end control from raw image RGB pixels.
>
> > This reviewer notes that the authors discussed CLIPort and the Nair et al. (2022) paper; however, these works were not benchmarked against. The paper does not present a clear argument against benchmarking against these approaches
>
> In terms of approach, BC-Z and MCIL are a lot more similar to DeL-TaCo and thus more easily comparable to our method than CLIPort or LoReL (Nair, et. al., 2022). Our method outperforms BC-Z and MCIL, as mentioned earlier.
>
> - CLIPort uses an entirely different action parametrization where an entire pick-place trajectory is parametrized by 2 points on a 2D plane (the pick and place points), whereas our approach predicts a 3D action at each timestep, which makes the action-prediction problem a lot harder.
> - LoReL is an RL approach that involves gathering a large dataset of human-labeled (demonstration, language) pairs and training a reward classifier to output the probability that two images show the completion of a language instruction. This is not directly comparable to our results because we use imitation learning instead of RL and thus do not utilize reward functions that can make use of this reward classifier. LoReL is also a model-based approach that involves learning a complex dynamics model, whereas ours is model-free.
>
> > It is unclear whether new theories or insights are gleaned (other than adding more input data can enhance performance by 5-9%).
>
> Reducing the main takeaway of DeL-TaCo to “adding more input data enhances performance” might be missing a few important points. We believe the insights of our work are much deeper than this for the following reasons:
> - __Our work presents a fairly unique and simple solution to a common challenge in multimodal learning.__ A common issue in multimodal learning is that the model only pays attention to the modality that is “easiest” to learn from, and the other modalities are almost completely ignored. Common prior approaches address this issue by randomly and selectively masking parts of each input modality stream so that the model is forced to pay attention to both modalities. We did not need to do any masking tricks in DeL-TaCo but were able to have the policy pay attention to the relevant information in both demo and language with a relatively simple and straightforward architecture and loss.
> - __The efficacy of multimodal task specification (demos + language) has been largely unexplored in robot learning.__ Previous work has not been able to get the combination of both demonstrations and language to help in generalization for robotic manipulation tasks over conditioning on a single modality alone. However, a combination of language and a single demonstration can be naturally and easily provided in most robotic learning contexts. We show how particular types of additional input can be more useful than others, i.e. language instructions are more useful than additional demonstrations. Specifically, in our problem setting, language can save the end-user from needing to provide 50 additional demonstrations. This has significant sample efficiency benefits, as it is easier to teach a robot a new skill by providing a single demo and language instruction to DeL-TaCo rather than 50 demos to a demo-only policy.

---

### Official Review · Reviewer_4Ugt · 2022-10-25

**Confidence:** 2
**Correctness:** 3
**Technical Novelty And Significance:** 3
**Empirical Novelty And Significance:** 3
**Recommendation:** 8

**Clarity, Quality, Novelty And Reproducibility:**

Clarity:
- very clear

Quality
- good quality

Novelty
- reasonably novel

Reproducibility
- high

**Strength And Weaknesses:**

Strengths:
- pretraining demonstration and language using contrastive loss is an interesting idea
- it's intuitive that conditioning on both improves performance, is a simple and elegant idea
- generalization results look decent
- the conclusion that language is worth 50 demos is useful, but this breaks down once you've 1000s of tasks

Weaknesses:
- tested on only sim, where data is very clean
- concerned that conditioning on one demo would make a policy rely on the spurious correlations of that demo : specific action situations, background, etc.
- given several demos of the same task, how would you sample the one to condition on? how do you scale this for large datasets


**Summary Of The Paper:**

Interesting paper, it uses language and demonstration conditioning to improve learning in an behaviour cloning set up for robot manipulation. The paper proposes:
1. a new architecture for training an eval conditioned on language and demonstrations together
2. some level of generalization with this

**Summary Of The Review:**

The paper proposes a method to condition on language and demo to solve robotic behaviour cloning. This is a decent set up in small-scale robot learning set ups. The idea is novel and simple. The claims of the paper are reasonably well supported within the data distribution used.

---

> ### Author Response · Authors · 2022-11-19
> **Response to Reviewer 4Ugt**
>
> Thank you for your review. We’re glad you found the paper interesting and will discuss some of the concerns and questions you have about the work.
>
> > concerned that conditioning on one demo would make a policy rely on the spurious correlations of that demo : specific action situations, background, etc.
>
> This is quite a valid concern. In fact, this is why we argue that providing language along with the single demo can help the policy understand what aspects of the demonstration are semantically meaningful and what aspects are unimportant. Our framework can also ameliorate some of these issues because it supports handling multiple demonstrations of a task, though we did not provide results in this setting. We expand on this in our answer to your next question.
>
> > given several demos of the same task, how would you sample the one to condition on? how do you scale this for large datasets
>
> We assume during deployment that the end user cannot afford to provide more than a few demonstrations. But in the scenario where there are multiple demonstrations, we can take an average over the $z_{demo}$ computed for each demo. This is what BC-Z does. From our experience, this boosts the performance of DeL-TaCo by quite a few percentage points. If we have a large number of demonstrations for each task, we could also finetune the policy on these demonstrations.
>
> To make this scalable for a large number of test tasks where we have many demonstrations per test task, future work can perhaps investigate how to select demonstrations that are most semantically similar to the environment the robot encounters, and then compute a weighted average over these demonstration embeddings to get an aggregated embedding for conditioning the policy.
>
> Please let us know if that answered your questions or if you have any additional questions. Thank you.

---

### Author Response · Authors · 2022-11-19
**Overall Response Sections 2-4 (Part 2/2) [For All Reviewers]**

# 2. Comparisons to Prior Work.
As a number of reviewers noted, this was lacking in the first version of our paper. In our update, we provide comparisons to two of the most relevant state-of-the-art methods that learn from both demonstrations/language (but only condition on one modality during test time): BC-Z [1] and MCIL [2].

We evaluate both prior works on Exp. Scenario A and get the following success rates (now added to Table 1 in the paper):
- BC-Z: 8.8%
- MCIL: 8.9%
- DeL-TaCo (ours; original): 19.9%

BC-Z does not perform well because it tries to regress $z_{demo}$ toward $z_{lang}$, and the policy is conditioned on $z_{demo}$. This suggests that BC-Z cannot perform better than using the “ground-truth” $z_{lang}$, and indeed, we find that BC-Z somewhat matches the performance of our language-only policy (~10%). In MCIL, there is no explicit task encoder loss term to directly shape the task embedding space meaningfully, lowering generalization performance compared to DeL-TaCo.

# 3. Extra motivation for demo+language task specification.
Some reviewers mentioned that the framework of using demonstrations and language is suboptimal compared to methods that only require, for instance, a single language instruction. We would like to provide additional motivation for robotic policies that leverage both demonstrations and language.

Humans learn new complex tasks, such as "how to make a tent," often by watching a video, which provides both a visual expert demonstration of how to do the task and also accompanying language instructions (via audio/speech) that help the learner follow along. Learning complex tasks is a lot harder if we only relied on a single modality (such as only demonstrations or only language instructions). Likewise, for a teacher, it is often harder to clearly specify new tasks with only one modality--imagine how difficult it would be to teach complex construction tasks if one could provide only demonstrations without speaking instructions, or could only provide language instructions without any visual guidance.

In this work, we argue and show that there exists robotic tasks complex enough where it is beneficial to provide both demonstrations and language instructions, as this is more efficient for both the end-user to specify and also the robot to learn from. We hope this provides a motivational perspective in addition to the focus on resolving ambiguities given in the introduction of the paper.

# 4. One-hot upper bounds for Exp. Scenario B.
Originally, we had provided one-hot oracle upper-bounds in Exp. Scenario A where we *fixed the number of expert trajectories per task* in the dataset. However, this led to large imbalances in the total number of training trajectories provided to the oracle upper-bounds versus all other methods. For instance, in Exp. Scenario B, there are ~250 train tasks and ~50 test tasks. This means that the oracle upper-bounds (trained only on the test tasks) were trained on 5x less total trajectories than the other methods, since we provided ~130 trajectories per task. Therefore, in our updated upper-bounds for both Exp. Scenarios A and B, we ensure that the oracles are trained with the *same amount of total trajectories in the training buffer.* This means that in Exp. Scenario B, for instance, the oracle is trained with 5x more trajectories per task since it is trained on 5x less tasks.

Exp. Scenario A:
- DeL-TaCo (ours; trained on ~130 trajs/task; original from Table 2): 19.9%
- One-hot Oracle (~130 trajs/task; originally shown in Table 1): 28.7%
- One-hot Oracle (~260 trajs/task; updated upper-bound): 47.9%

Exp. Scenario B:
- DeL-TaCo (ours; ~130 trajs/task; original from Table 2): 26.3%
- One-hot Oracle (~130 trajs/task): 15.6%
- One-hot Oracle (~650 trajs/task; updated upper-bound): 50.9%

These upper-bound results are updated in the paper (Tables 1 and 2) along with one-hot lower-bounds in Table 2.


# Concluding remarks.
Thanks again for all of your valuable feedback, suggestions, and questions. We will reply to each reviewer individually with more specifics. We would be happy to answer additional experimental/clarification questions during phase II of the review process.

# References.
[1] Eric Jang, Alex Irpan, Mohi Khansari, Daniel Kappler, Frederik Ebert, Corey Lynch, Sergey Levine, and Chelsea Finn. BC-Z: Zero-shot Task Generalization with Robotic Imitation Learning. In 5th Annual Conference on Robot Learning, 2021.

[2] Corey Lynch and Pierre Sermanet. Language conditioned imitation learning over unstructured data. Robotics: Science and Systems, 2021.

[3] Suraj Nair, Aravind Rajeswaran, Vikash Kumar, Chelsea Finn, and Abhinav Gupta. R3M: A Universal Visual Representation for Robot Manipulation. In 6th Annual Conference on Robot Learning, 2022.

[4] Oier Mees, Lukas Hermann, and Wolfram Burgard. What Matters in Language Conditioned Imitation
Learning. IEEE Robotics Automation and Letters, 2022.

---

### Author Response · Authors · 2022-11-19
**Overall Response Section 1 (Part 1/2) [For All Reviewers]**

We would like to thank all reviewers for their time and valuable suggestions. We have updated the paper, and all notable revisions are shown in $\textcolor{blue}{\text{blue text color}}$ in the PDF. We provide the following summary of our paper updates and additional comments.

# 1. Added Extensive Ablations.
All of the following ablation results are found in __Appendix F and Table 8 in the updated paper__. All numbers are success rates on Exp. Scenario A unless otherwise noted.

## A. Language Encoder Ablations
### A.i. Finetuning
Language-only:
- frozen DistilBERT (original from Table 1): 10.4%
- finetuned DistilBERT: 11.3%

DeL-TaCo:
- frozen DistilBERT (ours; original from Table 1): 19.9%
- finetuned DistilBERT: 11.8%

The DeL-TaCo policy is significantly harder to train jointly with a relatively deep language encoder. We suspect that training in stages, such as first freezing the encoder while learning the policy backbone, then freezing the policy backbone while finetuning the language encoder, may provide better results.

### A.ii. Comparing Language Models
Originally, we had only compared using two language encoders: CLIP and DistilBERT. In our updated paper, we additionally compare with DistilRoBERTa and miniLM. These language models were chosen because they are lightweight and are shown to perform among the best in a comparison of various language models for language-conditioned robotic policies done by [4]. In the following results, the language encoder is frozen during training.

Language-only:
- DistilBERT (original from Table 1): 10.4%
- DistilRoBERTa: 12.5%
- miniLM: 13.7%

DeL-TaCo:
- DistilBERT (ours; original from Table 1): 19.9%
- DistilRoBERTa: 16.6%
- miniLM: 20.9%

MiniLM slightly outperforms the other language models in both task embedding schemes. Additionally, DeL-TaCo consistently outperforms the language-only policy over the different language encoders.

### A.iii. Trainable MLP Head
In our updated paper, we provide results from training an MLP head after the frozen CLIP language encoder. This does not meaningfully impact performance.

DeL-TaCo with CLIP as demo + language encoder:
- Frozen CLIP (ours; original from Table 1): 15.3%
- Frozen CLIP + MLP head on language encoder: 15.8%

## B. Demo Encoder Ablations
Originally, we experimented with two demo encoders: pre-trained CLIP and trained-from-scratch CNN. In our updated paper, we also compare to using pre-trained R3M [3] (a ResNet-18 trained on human videos and shown to be helpful for downstream robotic tasks) with an added trainable MLP head on top.

DeL-TaCo:
- trained-from-scratch CNN (ours; original from Table 1): 19.9%
- R3M + trained MLP head: 13.0%

Similar to what we found for using pre-trained CLIP, using pre-trained R3M as the demo encoder did not perform better than a smaller CNN trained from scratch, perhaps because the simulated demonstrations, visually arranged in an array of images, are not within the distribution of the real-world-dominated training data of R3M, causing it to be unable to accurately distinguish between the simulated tasks in our benchmark.

## C. Task Encoder Loss Ablations
Originally, we presented results on using task encoders trained with contrastive learning. In our updates, we provide results using no task encoder loss term (only relying on the imitation learning loss, similar to MCIL), and using a cosine distance loss (similar to BC-Z).

DeL-TaCo, by task encoder loss term:
- Contrastive (ours; original from Table 1): 19.9%
- Cosine Distance: 12.2%
- No task encoder loss term: 12.4%

These results support our claim that for DeL-TaCo to outperform policies conditioned on only demo-only or language-only policies, $z_{emb}$ and $z_{lang}$ must contain complementary information. Cosine distance pushes $z_{emb}$ to approximate $z_{lang}$, making conditioning on both embeddings redundant. Having no task encoder loss results in less semantic structure in the task embedding space, which also hurts performance.

## D. Task Conditioning Architecture Ablations
Our original architecture consists of feeding $z_{demo}$ to the policy's FiLM layers while concatenating $z_{lang}$ to the image observation embeddings outputted by the policy's ResNet-18 backbone. In our updated paper, we provide additional comparisons: FiLM (demo + lang): concatenating both $z_{demo}$ and $z_{lang}$ and feeding this concatenation into the policy FiLM layers, and Concat (demo + lang): concatenating both $z_{demo}$ and $z_{lang}$ to the image observation embeddings without any FiLM layers.

DeL-TaCo, by task conditioning architecture:
- FiLM (demo), Concat (lang) (ours; original from Table 1): 19.9%
- FiLM (demo + lang): 17.4%
- Concat (demo + lang): 10.6%

This suggests that FiLM is crucial for multitask policies, and the embedding passed into the FiLM layers is fairly important.

___See the [following post](https://openreview.net/forum?id=4u42KCQxCn8&noteId=UD5GxSQtKb) for the final part of the Overall Response.___

---

### Decision · Program_Chairs · 2023-01-20

**Decision:**

Accept: poster

**Justification For Why Not Higher Score:**

The authors clearly put a lot of effort into responding to the reviews, which has lead to at least two reviewers revisiting their score.

It is worth noting that while one reviewer recommended 'accept', they listed their confidence level as being limited. As a result, the AC places less weight on that persons' review.

**Justification For Why Not Lower Score:**

As noted above, the AC would not object to the score being lowered in light of the reviewer who recommended 'accept' indicating their limited confidence in their decision. This, together with the fact that the AC personally knows (and trusts the opinion of) one of the reviewers make the AC less confident in the paper.

**Metareview: Summary, Strengths And Weaknesses:**

The paper considers the problem of learning multi-task policies from a combination of visual demonstrations and language utterances as complementary modalities. The paper proposes DeL-TaCo, a framework that conditions policies on language- and demonstration-based task embeddings that are aligned using contrastive learning. The framework is evaluated on a series of simulated robot manipulation tasks, with comparisons to single and multi-modality baselines. The results demonstrate the ability to learn multi-task manipulation policies that better generalize to novel tasks.

The paper was reviewed by four reviewers who agree that the idea of conditioning on both language and demonstrations as mutually beneficial sources of "supervision" is sensible and that the proposed framework is intuitive. The reviewers appreciate the analysis of the "value" of language input and find the paper to be very clear. The reviewers raised several concerns with the initial submission, notably the need for qualitative and quantitative comparisons to other single- and multi-modality baselines; additional experiments that explore greater test-time variations; and perceived inconsistencies between some of the claims made in the paper and the experimental results. It is clear from their response that the authors have put a significant amount of effort into responding to these concerns, which includes conducting several additional experiments as requested by the reviewers. The reviewers and AC appreciate these efforts, which have helped to clarify some of the initial reviewers' concerns initially raised, as reflected by two reviewers increasing their overall score.

This is an interesting and promising research direction. While the authors' response has certainly helped, the significance of the work is not yet fully clear. The paper would benefit from a more detailed analysis of the merits of language- and demonstration-based conditioning over their single modality counterparts (e.g., the additional experiments that investigate generalizability to test-time variation seem to suggest that DeL-TaCo is advantageous when language supervision is helpful, as measured by the language-only baseline, but can be inferior to a learning algorithm that only utilizes demonstrations). The authors are encouraged to consider a version of the paper that more thoroughly investigates the ability for the framework to take advantage of using language and demonstrations as complementary signals would provide a solid contribution to the robot learning community.

**Note From Pc:**

if the above contains the word "oral" or "spotlight" please see: "oral" presentation means -> notable-top-5% and "spotlight" means -> notable-top-25%. As stated in our emails, we are disassociating presentation type from AC recommendations

**Summary Of Ac-Reviewer Meeting:**

N/A